# REVISITING AND BENCHMARKING GRAPH AUTOEN-CODERS: A CONTRASTIVE LEARNING PERSPECTIVE

## ABSTRACT

Graph autoencoders (GAEs) are self-supervised learning models that can learn meaningful representations of graph-structured data by reconstructing the input graph from a low-dimensional latent space. Over the past few years, GAEs have gained significant attention in academia and industry. In particular, the recent advent of GAEs with masked autoencoding schemes marks a significant advancement in graph self-supervised learning research. While numerous GAEs have been proposed, the underlying mechanisms of GAEs are not well understood, and a comprehensive benchmark for GAEs is still lacking. In this work, we bridge the gap between GAEs and contrastive learning by establishing conceptual and methodological connections. We revisit the GAEs studied in previous works and demonstrate how contrastive learning principles can be applied to GAEs. Motivated by these insights, we introduce `lrGAE` (left-right `GAE`), a general and powerful GAE framework that leverages contrastive learning principles to learn meaningful representations. Our proposed `lrGAE` not only facilitates a deeper understanding of GAEs but also sets a new benchmark for GAEs across diverse graph-based learning tasks. The source code for `lrGAE`, including the baselines and all the code for reproducing the results, is publicly available at https://anonymous.4open.science/r/lrGAE/.

## 1 INTRODUCTION

In the last years, self-supervised learning (SSL) has emerged as a powerful learning paradigm for learning graph representations, approaching, and sometimes even surpassing, the performance of supervised counterparts on many downstream tasks Hjelm et al. (2019); van den Oord et al. (2018). Compared with supervised learning, self-supervised learning gets equal or even better performance with limited or no-labeled data which saves much annotation time and plenty of resources. In a nutshell, SSL purely makes use of rich unlabeled data via well-designed pretext tasks that exploit the underlying structure and patterns in the data. Most recent approaches are shaped by the design of pretext tasks and architectural design, which has led to two lines of research: contrastive and non-contrastive learning Garrido et al. (2023); Balestriero & LeCun (2022).

As one of the most successful and widespread SSL strategies, contrastive learning has first shown promising performance in vision representation learning Chen et al. (2020); Gao et al. (2021). It brings together embeddings of different views of the same image while pushing away the embeddings from different ones. Contrastive learning develops rapidly and has recently been applied to the graph learning domain because of the scarcity of graph datasets with labels. Contrastive learning on graphs (i.e., GCL) You et al. (2020) follows a similar framework to its counterparts in the vision domain, with the objective of maximizing the agreement between different graph augmentation views Wu et al. (2021); Li et al. (2024). Basically, contrastive views are designed as nodes, subgraphs, or mixtures of the two that are either similar (positive pairs) or dissimilar (negative pairs) Velickovic et al. (2019); Suresh et al. (2021); You et al. (2020); Xu et al. (2021).

On the other hand, non-contrastive learning approaches focus on capturing the generative aspects of graph data, with a promising line of research being graph autoencoders (GAEs) Kipf & Welling (2016a). Naïve GAEs adopt the edge-reconstruction principle to train the encoder, where edges of the input graph are expected to be reconstructed from hidden representations, thereby preserving the topological proximity and facilitating representation learning. Compared to contrastive methods,

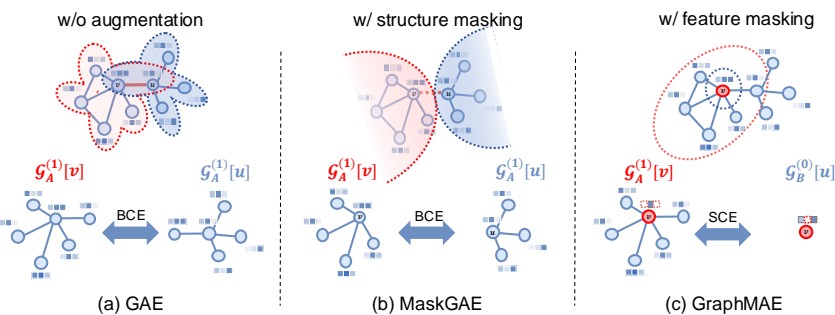

Figure 1: Technical comparison between GAE Kipf & Welling (2016a), MaskGAE Li et al. (2023b), and GraphMAE Hou et al. (2022) from a contrastive learning perspective.

self-supervised learning with GAEs is relatively less explored. This is particularly due to the fact that GAEs have been criticized for their limitations in capturing complex graph structures and potentially overemphasizing proximity information Velickovic et al. (2019); Li et al. (2023b). As a result, the rise of graph contrastive learning has led to a shift away from traditional GAEs, as researchers seek more effective approaches for graph self-supervised learning. So far, contrastive methods have long become a dominant SSL paradigm on graphs.

Until recently, GAEs based on masked autoencoding have renewed interest in the field of graph SSL. Masked autoencoding He et al. (2021) is a technique where a subset of the components (e.g., graph topology or node features) in the graph is randomly masked or corrupted Hou et al. (2022); Li et al. (2023b). By learning to reconstruct the graph from the partially masked input, GAEs can better capture the underlying structure and semantic information encoded in the graph. Masked GAEs offer improved performance and have the potential to capture the underlying graph properties, making them a preferred choice in many advanced graph learning tasks. Building on this momentum, research on (masked) GAEs has explored and become the new mainstream in graph self-supervised learning Shi et al. (2023); Wang et al. (2024); Tan et al. (2023); Hou et al. (2023).

While (masked) GAEs and contrastive approaches seem very different and have been described as such, we propose to take a closer look at the similarities between the two, both from a theoretical and empirical point of view. We argue that there exists a close relationship between GAEs and GCL. In this work, we revisit the GAEs studied in previous works and and closely look into GAEs from a contrastive learning perspective. To be specific, we demonstrate that

*GAEs, whether employing structure or feature reconstruction, with or without masked autoencoding, implicitly perform graph contrastive learning on two paired subgraph views.*

The equivalence between structure-based GAEs and graph contrastive learning was initially demonstrated by Li et al. (2023b) and further extended in our work by additionally taking feature-based GAEs into consideration. To support our claim, we provide an illustrative comparison of three representative GAEs from a contrastive learning perspective in Figure 1. Essentially, vanilla GAE Kipf & Welling (2016a) and MaskGAE Li et al. (2023b) are structure-based approaches that contrast a pair of connected nodes from their original and augmented structural views, respectively. On the other hand, GraphMAE Hou et al. (2022) is a feature-based approach that performs asymmetric graph contrastive learning by contrasting a node itself with its original and augmented subgraph views.

The above findings bridge the gap between graph contrastive and generative SSL methods, and further motivate us to unify GAEs into a contrastive architecture. In particular, we propose `lrGAE`, which formulates GAEs with five components: augmentation, contrastive views, encoder/decoder networks, contrastive loss, and negative samples. Among these, contrastive views are the core component that contributes to different aspects of the GAE architectures. We then relate popular GAEs to our architecture, highlighting the connections between them and further motivating the design of more effective contrastive-based GAE architectures. In this work, `lrGAE` is not only proposed as a general and powerful GAE framework but also a comprehensive GAE benchmark across different graph-based learning tasks.

Our contributions can be summarized as follows:

- **Connections between graph autoencoders and graph contrastive learning.** We establish a close connection between GAEs and GCLs over graphs. Our results demonstrate that GAEs implicitly contrast two paired subgraph views, and different designs of contrastive architectures facilitate the learning of GAEs. By pinpointing this equivalence, we consolidate our understanding of graph SSL methods.

- **Recipe for graph autoencoders.** Motivated by the equivalence, we make a significant effort to unify several state-of-the-art GAEs from a contrastive learning perspective and propose our recipe with four required and one optional step to design general and powerful GAEs. In particular, its steps are (1) augmentations, (2) contrastive views, (3) encoder/decoder networks, (4) contrastive loss, and dispensable (5) negative samples.

- **A unified framework and benchmark.** We present `lrGAE`, a unified and modular GAE framework that leverages contrastive learning principles and architecture design. `lrGAE`, as a benchmark, allows us to flexibly implement general yet powerful GAEs, with `lrGAE` ⑥⑦⑧, three advanced GAEs with asymmetric contrastive schemes as examples.

- **Experimental results and insights.** We conduct extensive benchmarking experiments on a series of graph-based learning tasks across a wide range of graph datasets, along with ablation studies that evaluate the contribution of several core components of `lrGAE`. Experimental results shed light on the effectiveness of contrastive views from the perspective of different downstream tasks.

To the best of our knowledge, `lrGAE` is the first work to explore contrastive learning principles and architecture design in the context of GAEs. As a general benchmark, we hope that `lrGAE` will contribute to a deeper understanding of GAEs, facilitate further research in the field, and enable the development of more effective graph self-supervised learning techniques.

## 2 RELATED WORK

To situate GAEs in a broader context, we discuss recent advances in graph self-supervised learning, including graph contrastive learning, graph autoencoders, and their recent masked variants.

**Graph contrastive learning.** Graph contrastive learning (GCL) is a general self-supervised learning paradigm excelling at capturing invariant information from diverse graph augmentation views. GCL has taken over the mainstream of self-supervised learning on graphs for years. Many works in this direction have recently flourished, with promising examples including DGI Velickovic et al. (2019), MVGRL Hassani & Ahmadi (2020), and GRACE Zhu et al. (2020). While most GCLs potentially suffer from scalability issues due to complex augmentation and sampling strategies, several efforts have been made to scale up GCL through augmentation-free paradigms (e.g., AFGRL Lee et al. (2022)), architecture simplification (e.g., BGRL Thakoor et al. (2021) and SGCL Sun et al. (2024)) or in-batch feature decorrelation (e.g., CCA-SSG Zhang et al. (2021)).

**Graph autoencoders.** Graph autoencoders (GAEs) are one such non-contrastive-based method that aim to learn meaningful representations by leveraging the graph reconstruction as the pretext task, i.e., reconstruct certain inputs within a given context. The pioneering works on GAEs can be traced back to GAE and VGAE Kipf & Welling (2016a), which utilize GNNs as the encoder and employ dot-product for link prediction decoding. Follow-up GAEs Li et al. (2023c); Pan et al. (2018); Wang et al. (2017); Hasanzadeh et al. (2019) mostly share a similar architecture, employing structure reconstruction or integrating both structure and feature reconstruction as their objectives. While GAEs typically excel in link-level tasks, they have been criticized for over-emphasizing proximity information at the expense of structural information and naturally underperform on node- and graph-level tasks pretrained on the graph reconstruction task Li et al. (2023b); Hou et al. (2022). This makes GAEs less preferable as a choice of graph self-supervised methods when compared to GCLs.

**Masked graph autoencoders.** Masked graph autoencoders are advanced GAEs that learn representations by reconstructing randomly masked patches from a graph input. There are two major lines of masked GAEs, with each line primarily focusing on reconstruction at either the feature level Hou et al. (2022; 2023); Shi et al. (2023); Wang et al. (2024) or the structure level Li et al. (2023b); Tan et al. (2023). Two seminal works include GraphMAE Hou et al. (2022) and MaskGAE Li et al. (2023b). GraphMAE introduces masked feature reconstruction on graphs as self-supervisions, while

MaskGAE focuses on graph structure reconstruction. The success of masked GAEs has spurred a surge of research in this area after the presentation. Later works seek to improve masked GAEs through the utilization of powerful encoders Zhang et al. (2022b), advanced masking strategies Shi et al. (2023), and regularization techniques Wang et al. (2024); Hou et al. (2023). However, a unified framework for GAEs and their masked alternatives is lacking, which motivates us to address this gap in our work.

## 3 GRAPH AUTOENCODERS: GENERATIVE YET CONTRASTIVE

In this section, we begin by revisiting graph contrastive learning (GCL) and graph autoencoders (GAEs). Then, we present our viewpoint on establishing the connections between GCLs and GAEs. Finally, we leave several remarks regarding the limitations of current GAEs.

**Graph contrastive learning.** Let $\mathcal{G} = (\mathcal{V}, \mathcal{E})$ be an attributed graph with $\mathcal{V} = \{v_i\}_1^N$ and $\mathcal{E} \subseteq \mathcal{V} \times \mathcal{V}$ the set of nodes and edges, respectively. $N$ is the number of nodes in the graph. GCLs commonly involve generating two augmented views, denoted as $\mathcal{G}_A$ and $\mathcal{G}_B$, and try to maximize the mutual information or the correspondence between two different views to train the encoder $f_\theta$. Here, $f_\theta$ represents an encoder network that maps the graph structure and node features simultaneously into a low-dimensional space. In this context, the goal of maximizing mutual information is achieved by minimizing the following objective:

$$\min_\theta \frac{1}{|\mathcal{V}|} \sum_{v \in \mathcal{V}} \mathcal{L}\left(\mathbf{Z}_A[v], \mathbf{Z}_B[v]\right), \tag{1}$$

where $\mathbf{Z}_A = f_\theta(\mathcal{G}_A), \mathbf{Z}_B = f_\theta(\mathcal{G}_B)$, and $\mathcal{L}$ refers to the contrastive loss, such as InfoNCE van den Oord et al. (2018). Typically, $f_\theta$ is a GNN network with receptive fields of $k$ (e.g., the depth of the network). We can further simplify Eq. 1 as follows:

$$\min_\theta \frac{1}{|\mathcal{V}|} \sum_{v \in \mathcal{V}} \mathcal{L}\left(\mathcal{G}_A^{(k)}[v], \mathcal{G}_B^{(k)}[v]\right), \tag{2}$$

where $\mathcal{G}_A^{(k)}[v]$ denotes the receptive fields of node $v$ in graph $\mathcal{G}$, with $f_\theta$ implicitly defined. In this way, we have a compact representation of GCL on its core component, i.e., the contrastive pair $\mathcal{G}_A^{(k)}[v]$ and $\mathcal{G}_B^{(k)}[v]$. In what follows, we will further generalize the representation of GAEs to adopt the form of this contrastive formulation.

**Graph autoencoders.** Technically, GAEs are encoding-decoding architectures that follows the graph-reconstruction principle as self-supervisions. The goal of GAEs is to reconstruct or decode graph components, such as edges or features, from hidden representations. A typical GAE consists of an encoder network $f_\theta$, similar to GCLs, which learns low-dimensional representations, as well as a decoder network $g_\phi$ that performs graph reconstruction pretext tasks. Here we first introduce the learning objective of the conventional GAE Kipf & Welling (2016a), which is to reconstruct the graph structure, following the form described in Li et al. (2023b):

$$\mathcal{L} = -\left(\frac{1}{|\mathcal{E}^+|} \sum_{(u,v) \in \mathcal{E}^+} \log g_\phi(\mathbf{Z}[u], \mathbf{Z}[v]) + \frac{1}{|\mathcal{E}^-|} \sum_{(u',v') \in \mathcal{E}^-} \log(1 - g_\phi(\mathbf{Z}[u'], \mathbf{Z}[v']))\right) \tag{3}$$

where $\mathcal{E}^+$ is a set of positive edges and is usually a subset of $\mathcal{E}$, i.e., $\mathcal{E}^+ \subseteq \mathcal{E}$. Correspondingly, $\mathcal{E}^-$ is a set of negative edges sampled from the graph and $\mathcal{E}^+ \cap \mathcal{E}^- = \emptyset$. $\mathbf{Z} = f_\theta(\mathcal{G})$ in which $f_\theta$ is a GNN network such as GCN Kipf & Welling (2016b) or GAT Veličković et al. (2018). $g_\phi$ is the decoder network, which can be a simple dot-product or an advanced neural network performed on the combined representations:

$$g_\phi(x, y) = x \cdot y \text{ or } g_\phi(x, y) = \text{MLP}_\phi(x \| y), \tag{4}$$

where $\text{MLP}_\phi$ denotes a multi-layer perceptron (MLP) network parameterized by $\phi$; $\|$ is the concatenate operation. Without loss of generality, we refer to GAEs that follow the form of Eq. 3 as *structure-based GAEs*.

Literature has shown that the structure-based GAEs might over-emphasize proximity information that is not always beneficial for self-supervised learning Velickovic et al. (2019); Li et al. (2023b);

Zhu et al. (2020), there are also works inheriting from VAE Kingma & Welling (2014) that seek to reconstruct the graph from its feature perspective:

$$\mathcal{L} = \frac{1}{|\mathcal{P}|} \sum_{(i,j)\in\mathcal{P}} (g_\phi(\mathbf{Z})[i,j] - \mathbf{X}[i,j])^2, \tag{5}$$

where $\mathcal{P}$ is the set of coordinates denoting the elements in the feature matrix to be reconstructed. $g_\phi$ is also a decoder network defined by an MLP on the input representations, i.e., $g_\phi(x) = \mathrm{MLP}_\phi(x)$. Similarly, we term GAEs following the form of Eq. 5 as *feature-based GAEs* or $\mathrm{GAE}_f$ for short.

**Connecting GAEs to GCLs.** Motivated by recent advancements in contrastive learning HaoChen et al. (2021); Zhang et al. (2022a); Li et al. (2023b), we present a viewpoint that places both structure-based GAEs and feature-based GAEs into an (approximate) contrastive learning framework. A few additional notations will be introduced: Let $\mathcal{S}(\mathcal{G}, k)$ denote the set of all rooted subgraphs of depth $k$ with root nodes ranging over the node set $\mathcal{V}$ of the graph $\mathcal{G}$. We define an *augmentation distribution* $\mathcal{A}(\cdot|s)$ to be a probability distribution that is supported on $\mathcal{S}(\mathcal{G}, k)$ conditioned on an element $s \in \mathcal{S}(\mathcal{G}, k)$. Let $p_\mathcal{G}$ be the empirical distribution of elements in $\mathcal{S}(\mathcal{G}, k)$. Following the seminal work of Wang & Isola (2020), we define two critical components of contrastive objectives as follows:

$$\mathcal{L}_{\text{alignment}} = \mathbb{E}_{s^+\sim\mathcal{A}(\cdot|s),s\sim p_\mathcal{G}} \left[ \ell_{\text{alignment}}(g_\theta(s), g_\theta(s^+)) \right] \tag{6}$$

$$\mathcal{L}_{\text{uniformity}} = \mathbb{E}_{(s,s_1^-,\ldots,s_M^-)\overset{\text{i.i.d}}{\sim} p_\mathcal{G}} \left[ \ell_{\text{uniformity}}(g_\theta(s), g_\theta(s_1^-), \ldots, g_\theta(s_M^-)) \right]. \tag{7}$$

The alignment loss $\mathcal{L}_{\text{alignment}}$ promotes the similarity of learned representations of positive sample pairs which are drawn according to certain augmentation mechanisms $\mathcal{A}$, while the uniformity loss $\mathcal{L}_{\text{uniformity}}$ encourages the diversity of learned representations to prevent collapsed solutions like constant mapping. To place GAEs in contrastive frameworks, the key step is to find suitably defined augmentation mechanisms as well as the forms of alignment and uniformity losses. We start from structure-based GAEs: Note that the first term in the right-hand side of Eq. 3 is equivalent to a log-loss (used as an alignment loss) under the *structural augmentation* $\mathcal{A}_S(\cdot|\mathcal{G}^k[v])$ defined as a uniform distribution over root-$k$ subgraphs corresponding to nodes adjacent to node $v$. However, the second term in the r.h.s of Eq. 3 cannot be cast into exact uniformity-type losses since the negative-sampling distribution is a biased approximation of $p_\mathcal{G}$. Therefore, structure-based GAEs could be regarded as an approximated contrastive learning defined by adjacency augmentations with a biased uniformity loss, an observation which has been revealed in previous work Li et al. (2023b).

The case of feature-based GAEs is more complicated than that of structure-based GAEs, since the augmentation is carried out in an *implicit manner*. We take inspiration from recent work Zhang et al. (2022a) by utilizing the following intuition: Two rooted subgraphs $s, s'$ are considered to be a positive pair if they are likely to share the same feature of their root nodes. To present the above intuition in a slightly more formal way, denote

$$\mathcal{M}_v^{(k)}(s|x) = \mathbb{P}\left[ \mathcal{G}^k[v] = s \Big| x_v = x \right] \tag{8}$$

as the conditional probability measuring the likelihood of some $x \in \mathcal{S}(\mathcal{G}, k)$ being generated under root feature $x$. Then we define the following feature augmentation mechanism:

$$\mathcal{A}_S\left(\mathcal{G}^{(k)}[u] \Big| \mathcal{G}^{(k)}[v]\right) \propto \mathbb{E}_x\left[ \mathcal{M}_u^{(k)}(\mathcal{G}^{(k)}[u]|x)\mathcal{M}_v^{(k)}(\mathcal{G}^{(k)}[v]|x) \right]. \tag{9}$$

Consequently, we have the following justification that feature-based GAE might be approximately regarded as performing an alignment-loss minimization procedure.

**Lemma 3.1.** *Under mild conditions, the GAE loss (Eq. 5) is lower bounded by an alignment loss which is induced by the inner product:*

$$\mathbb{E}\left[\mathcal{L}_{GAE}\right] \geq \frac{1}{2}\mathbb{E}_{s'\sim\mathcal{A}_F(\cdot|s),s\sim p_G}\left[\langle g_\theta(s), g_\theta(s')\rangle\right] + constant \tag{10}$$

The above lemma is a direct consequence of applying (Zhang et al., 2022a, Theorem 3.4) to our setup.

**Remark I: caveats of vanilla structure-based and feature-based GAEs.** Taking the perspective of (approximate) contrastive learning of GAEs would allow us to gain insights regarding their

deficiency. In particular, for structure-based GAEs, directly applying Eq. 3 would implicitly encode the information of a potentially large overlapped subgraph, thereby over-emphasizing proximity as elaborated in Li et al. (2023b). Alternatively, in feature-based GAEs, the vanilla objective Eq. 5 contains no components that account for uniformity regularization as in standard contrastive paradigms. Consequently, the GAE loss admits trivial shortcuts as optimal solutions (the constant map) which may deteriorate learning.

**Remark II: on expositions of theoretical connections and potential implications.** The expositions of theoretical connections between GAEs and GCLs were largely inspired by previously established theoretical insights. To the best of our knowledge, theoretical explorations of contrastive paradigms often rely on a given augmentation distribution and derive learnability or generalization results thereafter HaoChen et al. (2021); Parulekar et al. (2023). However, in lrGAE we are more focused on the design aspects of the augmentations, with preliminary theory work presented in Huang et al. (2021) but an extension to the field of graph learning remains highly non-trivial. Moreover, directly analyzing downstream generalization based on emerging self-supervised learning theories like Lee et al. (2021) is also challenging, as they typically rely on simple statistical models like linear models or topic models, which are not yet widely applicable in practical graph learning scenarios. Therefore, currently we believe a more detailed analysis tailored to the lrGAE framework is still beyond the scope of this paper but warrants extensive further studies.

## 4 LRGAE: DESIGN SPACE FOR GAEs FROM CONTRASTIVE LEARNING PERSPECTIVE

We have demonstrated that GAEs are such generative yet contrastive models, here we introduce lrGAE (left-right GAE), the first contrastive-based GAE architecture designed with the general purpose of learning powerful representations. Following the works in GCLs, we decompose the design space of lrGAE from five key dimensions: (1) augmentations, (2) contrastive views, (3) encoder/decoder networks, (4) contrastive loss, and dispensable (5) negative samples.

**Augmentations.** Graph augmentation is the first step in GCL, which generates multiple graph views from the input graph without affecting the semantic meaning You et al. (2020). These views are typically created by applying certain transformations, and the goal is to help the model learn robust and generalizable representations. In GCLs, the most prevalent augmentation techniques include node dropping, edge perturbation, and attribute/feature corruption You et al. (2020); Rong et al. (2020); Velickovic et al. (2019). However, as pointed out in § 3, the contrastive viewpoint of GAEs reveals the deficiency of augmentation mechanisms like information redundancy and collapsed solutions. To mitigate these issues, a recent line of work Li et al. (2023b); Hou et al. (2022) has been using a simple idea of masking to improve learning performance. Specifically, in structure-based GAEs, masking a certain proportion of edges effectively reduces information redundancy Li et al. (2023b). Meanwhile, masking the root node in feature-based GAEs Hou et al. (2022) can prevent trivial solutions (Zhang et al., 2022a, Theorem 3.6). Following the design space of GCL, we mainly consider lrGAE with node/edge/attribute masking as augmentations.

**Encoder/decoder networks.** An encoder maps the input graph into low-dimensional representations and is typically defined as a GNN network. Basically, the receptive fields of the encoder are determined by the depth of the GNN network. Meanwhile, the decoder network is regarded as a task-specific 'adapter', which maps augmented representations to another latent space where the contrastive loss is calculated for different pretext tasks, such as graph reconstruction. In most cases, GAEs employ an asymmetric design, where the decoder network is implemented as an MLP, although a GNN can also be used to enhance decoding and expand the receptive fields. However, using a GNN as the decoder may additionally introduce the oversmoothing issue. Multiple encoder and decoder can be adopted to learn diverse representations in different ways. Similar to GCLs, lrGAE do not apply any constraint on the encoder/decoder architecture.

**Contrastive views.** General GCLs typically follow a dual-branch architecture, which maximizes the agreement between two graph views (e.g., a graph and its augmentation). In contrastive learning, the choice of 'view' controls the information that the representation captures and influences the performance of the model. As depicted in Eq. 2, the contrastive views in GCLs can be simply denoted as $\mathcal{G}_A^{(k)}[v] \leftrightarrow \mathcal{G}_B^{(k)}[v]$. In a nutshell, in GCLs, the goal is to maximize the correspondence of the

$k$-hop neighborhood representations of a node $v$ and itself in two graph views, $\mathcal{G}_A$ and $\mathcal{G}_B$, where $k$ represents the receptive fields of the encoder network (e.g., a $k$-layer GNN). When it comes to GAEs, the same spirit could be followed where contrastive views are inherently determined by the graph structure. For example, in GAEs, adjacent nodes that have an explicit connection in the graph are considered positive pairs while non-connected nodes are treated as negative pairs. Following the practice of GCLs, we highlight three key components in the contrastive views of `lrGAE`, including *graph views, receptive fields, and node pairs*. Each of these components independently contributes to the performance of `lrGAE`. Therefore, we have $2^3$ cases of contrastive views by exhaustively enumerating all the possible combinations, as summarized in Table 1 with corresponding implementations in current works. Note that case ① is not applicable since it simply compares two original and identical samples as contrastive views, resulting in the loss becoming zero. Given that cases ②③④⑤ have been extensively explored and implemented in current works, cases ⑥⑦⑧ remain largely unexplored and have not received as much attention in the existing literature yet. We explore them within `lrGAE` framework to expand the design spaces.

Table 1: Eight cases of contrastive views in `lrGAE`.

| | Contrastive views: $\mathcal{G}_A^{(l)}[v] \leftrightarrow \mathcal{G}_B^{(r)}[u]$ | | | | | | | |
|---|---|---|---|---|---|---|---|---|
| | ① | ② | ③ | ④ | ⑤ | ⑥ | ⑦ | ⑧ |
| **Graph views** | $A = B$ | $A \neq B$ | $A = B$ | $A \neq B$ | $A = B$ | $A = B$ | $A \neq B$ | $A \neq B$ |
| **Receptive fields** | $l = r$ | $l = r$ | $l \neq r$ | $l \neq r$ | $l = r$ | $l \neq r$ | $l = r$ | $l \neq r$ |
| **Node pairs** | $v = u$ | $v = u$ | $v = u$ | $v = u$ | $v \neq u$ | $v \neq u$ | $v \neq u$ | $v \neq u$ |
| **Abbreviation** | AAllvv | ABllvv | AAlrvv | ABlrvv | AAllvu | AAlrvu | ABllvu | ABlrvu |
| **Implementations** | N/A | GCLs | GAE$_f$ $^{\dagger}$ | GraphMAE | MaskGAE | ? | ? | ? |

$^{\dagger}$ We refer to the feature-based variant (see Eq. 5) of GAE here.

**Contrastive loss.** Contrastive loss or objective is designed to train models to distinguish between similar (positive) and dissimilar (negative) pairs of graph views according to different pretext tasks. The main idea in vanilla GCLs is to bring the representations of augmented views of the graph closer together while pushing the representations of different graphs further apart. By effectively contrasting positive and negative pairs, the model learns to capture the essential structural and feature information of the graphs, leading to improved performance on various downstream tasks. For GAEs, whose core idea is to perform reconstruction over graph structure or node features, the learning objectives involve binary cross-entropy (BCE) loss or regression loss mean squared error (MSE), respectively. As discussed in § 3, both losses follow the same contrastive learning philosophy. As we have established a strong connection between GAEs and GCLs, we can further advance GAEs with various well-established contrastive learning objectives beyond simple BCE or MSE in `lrGAE`.

**Negative samples.** Negative samples are instances that are dissimilar to the positive samples in contrastive learning, which act as a reference for the learning model to distinguish between meaningful patterns and random noise. Conventional GCLs largely rely on negative samples to learn meaningful representations and capture important patterns in the data. However, this approach comes with a trade-off in learning efficiency, and the quality of negative samples is not always guaranteed. Since graph structure reconstruction in GAEs is inherently tackled as a binary link classification task, negative links or edges are necessary to prevent model collapse and encourage learning meaningful representations. In contrast, feature reconstruction focuses on recovering node-level or graph-level features without the need for negative samples since the task revolves around reproducing the original feature representations. Recent advances have shown that GCLs can also be effective even without explicit negative samples Thakoor et al. (2021); Zhang et al. (2021); Sun et al. (2024). This opens up the possibility of eliminating negative samples in `lrGAE` by properly incorporating several techniques from GCL such as stop-gradient or asymmetric networks.

**Adapting current GAEs to `lrGAE` framework.** Since we have established the close connections between GAEs and GCLs, we are now able to relate popular GAEs to our proposed `lrGAE` framework. Table 2 summarizes several advanced GAEs as well as conventional GCLs within our proposed contrastive framework across four dimensions. We omit the encoder/decoder networks here as most methods share a similar network architecture. GAEs are tailored by their asymmetric contrastive views design, either on graph views or receptive fields. In addition, most existing GAEs follow the conventional learning paradigms, i.e., regression loss (MSE or SCE Hou et al. (2022)) for

Table 2: Comparison of different (masked) GAEs that fall into our contrastive framework `lrGAE`. GCL is also included in the table for reference. Here $k$ is the depth/receptive fields of the encoder/decoder networks.

| | Augmentation | Contrastive views | | Contrastive Loss | Negative samples |
|---|---|---|---|---|---|
| **GCL** | Edge/feature corruption, ... | $\mathcal{G}_A^{(k)}[v]$ | $\leftrightarrow\ \mathcal{G}_B^{(k)}[v]$ | InfoNCE, SimCSE, ... | ✓† |
| **GAE$_f$** Kipf & Welling (2016a) | - | $\mathcal{G}^{(k)}[v]$ | $\leftrightarrow\ \mathcal{G}^{(0)}[v]$ | MSE | ✗ |
| **GAE** Kipf & Welling (2016a) | - | $\mathcal{G}^{(k)}[v]$ | $\leftrightarrow\ \mathcal{G}^{(k)}[u]$ | BCE | ✓ |
| **MaskGAE** Li et al. (2023b) | Edge/path masking | $\mathcal{G}_A^{(k)}[v]$ | $\leftrightarrow\ \mathcal{G}_A^{(k)}[u]$ | BCE | ✓ |
| **S2GAE** Tan et al. (2023) | Directed edge masking | $\mathcal{G}_A^{(k)}[v]$ | $\leftrightarrow\ \mathcal{G}_A^{(k)}[u]$ | BCE | ✓ |
| **GraphMAE** Hou et al. (2022) | Feature masking | $\mathcal{G}_A^{(k)}[v]$ | $\leftrightarrow\ \mathcal{G}_B^{(0)}[v]$ | MSE/SCE | ✗ |
| **GraphMAE2** Hou et al. (2023) | Feature masking | $\mathcal{G}_A^{(k)}[v]$ | $\leftrightarrow\ \mathcal{G}_B^{(0)}[v]\ \cup\ \mathcal{G}^{(k)}[v]$ | MSE/SCE | ✗ |
| **AUG-MAE** Wang et al. (2024) | Adaptive feature masking | $\mathcal{G}_A^{(k)}[v]$ | $\leftrightarrow\ \mathcal{G}_B^{(0)}[v]$ | MSE/SCE | ✗ |
| **GiGaMAE** Shi et al. (2023) | Edge & feature masking | $\mathcal{G}_A^{(k)}[v]$ | $\leftrightarrow\ \mathcal{G}_B^{(0)}[v]$ | InfoNCE | ✗ |

† Note that there are a set of works in GCLs that do not require negative samples Thakoor et al. (2021); Zhang et al. (2021); Sun et al. (2024).

feature-based reconstruction, while classification loss is used for structure-based reconstruction. Until recently, only GiGaMAE has attempted to adapt other GCL losses (e.g., InfoNCE Tschannen et al. (2020); Chen et al. (2020)) to GAEs, leaving the potential of GAEs with more powerful GCL techniques largely unexplored.

## 5 EXPERIMENTS

In this section, we perform extensive experiments over several graph learning tasks to benchmark the performance of GAEs and three variants of `lrGAE`. Specifically, the experiments are conducted on seven graph datasets, including Cora, CiteSeer, PubMed Sen et al. (2008), Photo, Computers Shchur et al. (2018), CS and Physics Shchur et al. (2018). The comparison baselines include vanilla GAE Kipf & Welling (2016a) and its variant GAE$_f$, as well as masked GAEs, i.e., MaskGAE Li et al. (2023b), S2GAE Tan et al. (2023), GraphMAE Hou et al. (2022), GraphMAE2 Hou et al. (2023), and AUG-MAE Wang et al. (2024). Since `lrGAE` ②③④⑤ have their corresponding implementations (see Table 1), we implement `lrGAE` ⑥⑦⑧ in experiments for comparison. Due to space limitation, we kindly refer readers to Appendix for more details of datasets, baseline methods, evaluation settings, and implementation details of `lrGAE`.

### 5.1 MAIN RESULTS

**Link prediction.** Table 3 presents the link prediction results of GAEs in three citation graphs. We follow exactly the experimental settings in Kipf & Welling (2016a) and report the averaged AUC and AP across 10 runs. For feature-based GAEs that without trained with a structural decoder, we use dot-product operation over the learned node representations to perform link prediction. As can be observed, masked GAEs are more powerful than vanilla GAEs. The results are in line with the conclusions of previous works Li et al. (2023b); Hou et al. (2022); Wang et al. (2024) that augmentations such as masking can greatly benefit self-supervised learning. In addition, structure-based GAEs, i.e., GAE, S2GAE, MaskGAE, and `lrGAE` ⑥⑦⑧, generally achieve better performance on all datasets. By contrast, feature-based GAEs underperform in such a link prediction task due to the gap between the pretext task (regression) and the downstream task (classification) of feature-based GAEs. This highlights the importance of generality of pretext task during self-supervised learning. It is worth noting that although GiGaMAE adopts a feature-reconstruction objective, it also achieves comparable performance with structure-based GAEs. The possible reason might be that the contrastive learning loss offers better generalization than regression loss (e.g., MSE and SCE Hou et al. (2022)) for GAEs.

**Node classification.** To provide a comprehensive benchmark, we conduct node classification task on seven graph benchmarks commonly used in literature. For the Cora, CiteSeer, and PubMed datasets, we utilize the publicly available splits as described in previous works Kipf & Welling (2016b); Li et al. (2023b). For the remaining datasets, we follow the recommended 8:1:1 train/validation/test random splits, as advocated by Shchur et al. (2018). Table 4 present the node classification results across

Table 3: Link prediction results (%) on three citation networks. In each column, the **boldfaced** score denotes the best result and the underlined score represents the second-best result.

| | Cora | | CiteSeer | | PubMed | |
|---|---|---|---|---|---|---|
| | AUC | AP | AUC | AP | AUC | AP |
| $GAE_f$ | $75.0_{\pm1.2}$ | $74.3_{\pm0.9}$ | $70.6_{\pm2.4}$ | $70.8_{\pm2.1}$ | $82.2_{\pm0.5}$ | $81.3_{\pm0.4}$ |
| GraphMAE | $93.9_{\pm0.4}$ | $94.1_{\pm0.2}$ | $93.8_{\pm0.5}$ | $94.9_{\pm0.5}$ | $95.9_{\pm0.3}$ | $95.5_{\pm0.2}$ |
| GraphMAE2 | $94.1_{\pm0.5}$ | $94.3_{\pm0.4}$ | $91.9_{\pm0.5}$ | $93.4_{\pm0.6}$ | $95.6_{\pm0.1}$ | $95.2_{\pm0.3}$ |
| AUG-MAE | $95.9_{\pm0.6}$ | $96.2_{\pm0.3}$ | $94.7_{\pm0.2}$ | $95.8_{\pm0.3}$ | $94.5_{\pm0.2}$ | $94.0_{\pm0.1}$ |
| GiGaMAE | $93.5_{\pm0.5}$ | $94.4_{\pm0.7}$ | $97.5_{\pm0.4}$ | $97.2_{\pm0.1}$ | $97.5_{\pm0.3}$ | $97.3_{\pm0.4}$ |
| GAE | $92.5_{\pm0.8}$ | $93.7_{\pm0.6}$ | $88.4_{\pm1.3}$ | $90.5_{\pm1.2}$ | $98.0_{\pm0.2}$ | $98.1_{\pm0.3}$ |
| S2GAE | $94.5_{\pm0.5}$ | $93.8_{\pm0.4}$ | $94.0_{\pm0.2}$ | $95.3_{\pm0.3}$ | $98.3_{\pm0.1}$ | $98.2_{\pm0.4}$ |
| MaskGAE | $96.8_{\pm0.2}$ | $97.0_{\pm0.3}$ | $97.6_{\pm0.1}$ | $97.9_{\pm0.1}$ | $98.7_{\pm0.1}$ | $98.8_{\pm0.0}$ |
| lrGAE ⑥ | $96.3_{\pm0.6}$ | $96.1_{\pm0.7}$ | $96.5_{\pm0.2}$ | $96.4_{\pm0.2}$ | $98.1_{\pm0.1}$ | $97.7_{\pm0.3}$ |
| lrGAE ⑦ | $96.4_{\pm0.8}$ | $97.2_{\pm0.4}$ | $97.7_{\pm0.3}$ | $97.2_{\pm0.4}$ | $98.9_{\pm0.3}$ | $98.8_{\pm0.1}$ |
| lrGAE ⑧ | $96.3_{\pm0.5}$ | $96.2_{\pm0.5}$ | $97.1_{\pm0.1}$ | $97.2_{\pm0.5}$ | $98.8_{\pm0.1}$ | $98.4_{\pm0.2}$ |

Table 4: Node classification accuracy (%) on seven benchmark datasets. `OOM`: out-of-memory on an NVIDIA 3090ti GPU with 24GB memory.

| | Cora | CiteSeer | PubMed | Photo | Computers | CS | Physics |
|---|---|---|---|---|---|---|---|
| $GAE_f$ | $57.7_{\pm1.3}$ | $47.5_{\pm1.3}$ | $60.0_{\pm0.6}$ | $80.1_{\pm0.2}$ | $70.9_{\pm0.3}$ | $78.0_{\pm0.4}$ | $55.6_{\pm0.7}$ |
| GraphMAE | $84.5_{\pm1.1}$ | $72.5_{\pm0.9}$ | $81.0_{\pm0.8}$ | $93.3_{\pm0.2}$ | $89.8_{\pm0.2}$ | $92.7_{\pm0.4}$ | $95.1_{\pm0.2}$ |
| GraphMAE2 | $83.7_{\pm2.1}$ | $71.7_{\pm1.5}$ | $80.4_{\pm1.0}$ | $93.2_{\pm0.5}$ | $89.6_{\pm0.4}$ | $92.7_{\pm0.5}$ | $94.9_{\pm0.1}$ |
| AUG-MAE | $83.9_{\pm1.4}$ | $73.1_{\pm2.1}$ | $80.3_{\pm1.5}$ | $93.4_{\pm0.8}$ | $89.6_{\pm0.5}$ | OOM | OOM |
| GiGaMAE | $83.2_{\pm1.5}$ | $69.8_{\pm1.8}$ | $82.5_{\pm1.5}$ | $93.5_{\pm0.5}$ | $89.7_{\pm0.7}$ | $92.4_{\pm0.7}$ | OOM |
| GAE | $79.8_{\pm2.5}$ | $64.9_{\pm2.7}$ | $76.2_{\pm1.9}$ | $89.9_{\pm0.5}$ | $79.2_{\pm0.3}$ | $92.0_{\pm0.5}$ | $95.2_{\pm0.1}$ |
| S2GAE | $84.1_{\pm1.4}$ | $72.3_{\pm1.5}$ | $81.5_{\pm0.8}$ | $93.1_{\pm0.3}$ | $89.1_{\pm0.4}$ | $92.6_{\pm0.1}$ | $95.5_{\pm0.2}$ |
| MaskGAE | $84.4_{\pm1.5}$ | $72.8_{\pm0.7}$ | $82.9_{\pm0.4}$ | $93.0_{\pm0.1}$ | $89.5_{\pm0.1}$ | $92.9_{\pm0.1}$ | $95.6_{\pm0.2}$ |
| lrGAE ⑥ | $84.0_{\pm1.5}$ | $73.0_{\pm1.8}$ | $81.9_{\pm0.5}$ | $93.5_{\pm0.3}$ | $89.1_{\pm0.3}$ | $92.9_{\pm0.4}$ | $95.4_{\pm0.1}$ |
| lrGAE ⑦ | $84.2_{\pm1.8}$ | $72.5_{\pm1.1}$ | $81.1_{\pm0.8}$ | $93.4_{\pm0.5}$ | $89.8_{\pm0.2}$ | $92.8_{\pm0.2}$ | $95.7_{\pm0.2}$ |
| lrGAE ⑧ | $84.5_{\pm1.4}$ | $72.4_{\pm1.7}$ | $82.5_{\pm1.0}$ | $93.2_{\pm0.4}$ | $89.3_{\pm0.1}$ | $93.1_{\pm0.2}$ | $95.8_{\pm0.1}$ |

ten runs. It is observed that masked GAEs yield equally well performance, particularly in Photo, Computers, and CS. However, most feature-based GAEs suffer from scalability issues in Physics, which has a large feature dimensionality. This reveals the potential limitation of feature-based GAEs, which are less flexible and scalable compared to structure-based GAEs. We can also see that, lrGAE ⑥⑦⑧, three variants of GAEs with asymmetric graph views or receptive fields, have matched or even outperformed state-of-the-art performance in all cases. lrGAE unleashes the power of GAEs with GCL principles and provides deep insights to facilitate self-supervised learning.

Due to space limitations, we defer additional experimental results as well as ablation analysis in Appendix D and E.

## 6 CONCLUSION

In this work, we first show that GAEs are not only generative but also contrastive self-supervised models that contrast two paired subgraph views. Built upon the equivalence between GAEs and GCLs, we present lrGAE, a comprehensive GAE benchmark that leverages the contrastive learning principles to unify existing GAE approaches. Our extensive benchmarking experiments across diverse graph datasets and tasks, coupled with detailed ablation studies, have provided valuable insights into the effectiveness of contrastive views and the contributions of core components within lrGAE. Our work is setting the foundation for a unified architecture of graph self-supervised learning, with modular and scalable GAEs and a broader understanding of the role of GAEs with different contrastive designs. So far, since GAEs and their masked variants are a promising research direction, lrGAE should be easily extended to be compatible with newly proposed approaches in the future.

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

## A LrGAE FRAMEWORK AND IMPLEMENTATIONS

`lrGAE` is introduced as a versatile and comprehensive framework that offers flexibility in implementing powerful GAEs through customization of augmentations, contrastive views, encoder/decoder networks, and contrastive loss. In particular, the design of *contrastive views* and the corresponding learning loss play a crucial role in addressing various graph-based learning tasks efficiently and effectively. For example, different tasks may require different types of contrastive views, such as node-level views, link-level views, or even more complex views that incorporate both local and global graph information. Here we provide the PyTorch Paszke et al. (2019) style pseudocode for the implementation of `lrGAE` in Algorithm 1.

---

**Algorithm 1** PyTorch Paszke et al. (2019) style pseudocode for `lrGAE`.

```
# g: input graph
# gA, gB: graph views
# l, r: receptive fields
# v, u: node pairs

# graph augmentation
gA, gB = augmentation(g)

# encoding
left = encoder(gA)[l]
right = encoder(gB)[r]

# decoding
viewV = decoder(left[v])
viewU = decoder(right[u])

# contrasting
loss = loss_fn(viewV, viewU)
```

---

To further showcase the versatility and broad applicability of `lrGAE`, we list seven variants of `lrGAE` with different contrastive views below:

- `lrGAE` ②-$ABllvv$: This variant can be simply implemented with the same architecture as naïve GCLs Zhu et al. (2020); You et al. (2020), which contrast two augmentation views ($\mathcal{G}_A$ and $\mathcal{G}_B$) for each node ($v$) in its $l$-hop neighborhood (i.e., receptive field).

- `lrGAE` ③-$AAlrvv$: This is a variant of `lrGAE` ②-$ABllvv$ that contrasts the node representations from different layers of the encoder, i.e., the receptive fields ($l$ and $r$), which shares similar philosophy of local-to-global graph contrastive learning Velickovic et al. (2019). The most typical example is GAE Kipf & Welling (2016a) with feature reconstruction as its learning objective (i.e., GAE$_f$).

- `lrGAE` ④-$ABlrvv$: This is the core idea of GraphMAE Hou et al. (2022). By combining the success of GCLs and GAEs, we are able to develop this variant that incorporates different graph views ($\mathcal{G}_A$ and $\mathcal{G}_B$) and receptive fields ($l$ and $r$) of a node $v$.

- `lrGAE` ⑤-$AAllvu$: This is the learning paradigm of vanilla GAE and MaskGAE Li et al. (2023b), which contrasts the two subgraph pairs of nodes associated with an edge ($v, u$). Note that the graph view $\mathcal{G}_A$ can be the original graph $\mathcal{G}$ or the augmented/masked graph, leading to the implementations of GAE or MaskGAE.

- `lrGAE` ⑥-$AAlrvu$, `lrGAE` ⑦-$ABllvu$, and `lrGAE` ⑧-$ABlrvu$: The variants that do not have the exact implementation yet. Following the architecture of `lrGAE` ⑤-$AAllvu$, we can vary the receptive fields ($l$ and $r$), augmentation views ($\mathcal{G}_A$ and $\mathcal{G}_B$), or even both to perform graph contrastive learning.learning.

As shown in Table 1, `lrGAE` ②③④⑤ have implementations proposed in previous works Zhu et al. (2020); You et al. (2020); Kipf & Welling (2016a); Hou et al. (2022); Li et al. (2023b), we mainly provide the empirical results of `lrGAE` ⑥⑦⑧ in our experiments. The above variants cover the possible implementations of GAEs, further demonstrating the flexibility and adaptability of `lrGAE`.

Figure 2: Illustration of seven possible cases of `lrGAE`. We vary the graph views ($A$ and $B$), receptive fields ($l$ and $r$), as well as node pairs ($v$ and $u$) to implement different variants of `lrGAE` with different contrastive views.

This allows researchers to choose and customize the appropriate contrastive views based on their specific tasks and objectives. We present the illustration of the seven variants of `lrGAE` in Figure 2.

# B    DISCUSSIONS

Table 5: Time and space complexity of `lrGAE` framework. $|E|$ is the number of edges and $|V|$ is the number of nodes, $L$ is number of layers, $d$ is number of features. For simplicity, we assume the number of features is fixed across all layers. For the mini-batch training setting, $B$ is the batch size and $r$ is the number of sampled neighbors per node.

|  | Full-batch training (e.g., GCN) | Mini-batch training (e.g., GraphSAGE) |
|---|---|---|
| Time complexity | $\mathcal{O}(L|\mathcal{E}|d + L|\mathcal{V}|d^2)$ | $\mathcal{O}(r^L|\mathcal{V}|d^2)$ |
| Memory complexity | $\mathcal{O}(L|\mathcal{V}|d + Ld^2)$ | $\mathcal{O}(Br^L d + Ld^2)$ |

**Time and space complexity.**  We briefly discuss the time and space complexity of our proposed `lrGAE` framework. `lrGAE` is a standard GAE framework consisting of one encoder and one decoder network. As decoder networks are typically simple feed-forward networks (e.g., MLPs) involved with dense matrix computation, the major bottleneck arises from the message passing and aggregation of encoder networks (e.g., GCN Kipf & Welling (2016b)). In particular, for an $L$ layer encoder network, the time complexity is related to the graph size and the dimension of features and hidden representations, about $\mathcal{O}(L|\mathcal{E}|d + L|\mathcal{V}|d^2)$. By incorporating mini-batch training Hamilton et al. (2017), the time complexity for each sampled subgraph can be reduced to $\mathcal{O}(r^L|\mathcal{V}|d^2)$, where $r$ represents the neighborhood size shared by each hop. As for the space complexity, the major bottleneck also lies in the graph encoder network, which is $\mathcal{O}(L|\mathcal{V}|d + Ld^2)$ and $\mathcal{O}(Br^L d + Ld^2)$ for full-batch and mini-batch training, respectively. Here, $B$ denotes the batch size. Overall, the time and space complexity of `lrGAE` framework are guaranteed and can easily scale to larger graphs, showcasing its scalability and generality.

**Limitations and outlook.**  This work primarily presents initial benchmarks and baselines for GAEs, with a main focus on masked autoencoding based methods. Our work might potentially suffer

from some limitations: (i) Graph augmentation, such as masking, is a core operation of `lrGAE`. However, similar to existing augmentation-based GCL methods, performing masking on the graph structure could harm the semantic meaning of some graphs, such as biochemical molecules. Zhao et al. (2021). (ii) The theoretical results of `lrGAE` are mainly based on the homophily assumption, a basic assumption in the literature of graph-based representation learning. However, such an assumption may not always hold in heterophilic graphs, where the labels of linked nodes are likely to differ. This could potentially limit the applicability of `lrGAE` on heterophilic graphs. Our current analysis is based solely on existing datasets; hence, we plan to enhance the scope of `lrGAE` by acquiring and incorporating more diverse datasets and domains. Furthermore, while this paper primarily conducts an empirical investigation on GAEs, advancing the theoretical framework of `lrGAE` will be crucial for its development and understanding.

**Broader impact.** In this work, we introduce `lrGAE`, a general GAE framework that integrates and extends GAEs with powerful graph contrastive learning principles. `lrGAE` is also proposed as a comprehensive benchmark with the establishment of a standardized evaluation protocol to ensure fair and consistent comparisons in this line of research. The `lrGAE` benchmark encourages the development of new algorithms and techniques for graph representation learning. This can lead to improved methods for analyzing and understanding complex network structures, which can have wide-ranging positive impacts in various fields such as social network analysis, bioinformatics, and recommendation systems. While a standardized benchmark can promote comparability, it may also inadvertently discourage diversity in research approaches and lead to an overemphasis on performance metrics. The focus on benchmark performance metrics may inadvertently discourage the exploration of novel or unconventional methods that may offer unique insights but perform less well on standardized metrics. This could limit the potential for innovative approaches that could have a significant social impact but may not fit within the benchmark's evaluation framework.

Table 6: Statistics of dataset for three fundamental graph learning tasks. N: node classification; L: link prediction; C: graph clustering.

| Dataset | #Nodes | # Edges | #Features | #Classes | Tasks |
|---------|--------|---------|-----------|----------|-------|
| **Cora** | 2,708 | 10,556 | 1,433 | 7 | N/L/C |
| **CiteSeer** | 3,327 | 9,104 | 3,703 | 6 | N/L/C |
| **PubMed** | 19,717 | 88,648 | 500 | 3 | N/L/C |
| **Photo** | 7,650 | 238,162 | 745 | 8 | N/C |
| **Computer** | 13,752 | 491,722 | 767 | 10 | N/C |
| **CS** | 18,333 | 163,788 | 6,805 | 15 | N/C |
| **Physics** | 34,493 | 495,924 | 8,415 | 5 | N/C |

Table 7: Statistics of dataset for graph classification task (G).

| Dataset | Avg. #nodes | Avg. #edges | #Graphs | #Features | #Classes | Task |
|---------|-------------|-------------|---------|-----------|----------|------|
| **IMDB-B** | 19.8 | 193.1 | 1,000 | 0 | 2 | G |
| **IMDB-M** | 13.0 | 131.9 | 1,500 | 0 | 3 | G |
| **PROTEINS** | 39.1 | 145.6 | 1,113 | 4 | 2 | G |
| **COLLAB** | 74.5 | 4914.4 | 5,000 | 0 | 3 | G |
| **MUTAG** | 17.9 | 39.6 | 188 | 7 | 2 | G |
| **REDDIT-B** | 429.6 | 995.5 | 2,000 | 0 | 2 | G |
| **NCI1** | 29.9 | 64.6 | 4,110 | 37 | 2 | G |

Table 8: Statistics of dataset for heterogeneous node classification task (HN).

| Dataset | #Nodes | #Ndges | #Node types | #Edge types | #Classes | Task |
|---------|--------|--------|-------------|-------------|----------|------|
| **DBLP** | 26,128 | 239,566 | 4 | 6 | 4 | HN |
| **ACM** | 10,942 | 547,872 | 4 | 8 | 3 | HN |
| **IMDB** | 21,420 | 86,642 | 4 | 6 | 5 | HN |

## C REPRODUCIBILITY

All of `lrGAE`'s experimental results are highly reproducible. We provide more detailed information on the following aspects to ensure the reproducibility of the experiments.

**Datasets.** We conduct experiments on seven graph benchmark datasets, including three citation networks, i.e., Cora, CiteSeer, and PubMed Sen et al. (2008), two Amazon co-purchase graphs, i.e., Photo and Computer Shchur et al. (2018), two co-author graphs, i.e., CS and Physics Shchur et al. (2018). The above datasets are used for the tasks of node classification, link prediction, and graph clustering. For the graph classification task, we perform experiments on the following seven datasets: IMDB-B, IMDB-M, PROTEINS, COLLAB, MUTAG, REDDIT-B, and NCI1 Yanardag & Vishwanathan (2015). Each dataset consists of a set of graphs, with each graph associated with a corresponding label. We also incorporate three heterogeneous graph datasets, i.e., DBLP, ACM, and IMDB Lv et al. (2021); Li et al. (2023a), for experiments on heterogeneous node classification to showcase the generality and versatility of our `lrGAE` framework. All datasets used throughout experiments are publicly available at PyTorch Geometric Fey & Lenssen (2019). Detailed information about datasets are summarized in Table 6, Table 7 and Table 8.

**Baselines.** In line with the focus of this work, we benchmark several GAEs in different graph learning tasks, including vanilla GAE Kipf & Welling (2016a) and its variant $GAE_f$, as well as masked GAEs, i.e., MaskGAE Li et al. (2023b), S2GAE Tan et al. (2023), GraphMAE Hou et al. (2022), GraphMAE2 Hou et al. (2023), and AUG-MAE Wang et al. (2024). Among the baselines, GAE, MaskGAE, and S2GAE adopt structure reconstruction as their learning objective, while GraphMAE, GraphMAE2, and AUG-MAE adopt feature reconstruction as their learning objective. We implement baselines with PyTorch Paszke et al. (2019) and PyTorch Geometric Fey & Lenssen (2019), which are open-source software released under BSD-style [1] and MIT [2] license, respectively. For feature-based GAEs ($GAE_f$, GraphMAE, GraphMAE2, AUG-MAE, GiGaMAE), we employ the GAT Veličković et al. (2018) architecture and scaled cosine error (SCE) as the encoder network and learning objective. On the other hand, for structure-based GAEs (GAE, MaskGAE, S2GAE), we utilize the GCN Kipf & Welling (2016b) architecture and binary cross-entropy as the encoder network and learning objective, respectively.

**Implementation details.** To align with the baseline implementations, we abstract the contrastive learning principles of GAEs and implement `lrGAE` with PyTorch and PyTorch Geometric as well. Specifically, we have seven basic variants of `lrGAE` with different contrastive views. Specifically, we refer `lrGAE` ②③④ as feature-based variants while `lrGAE` ⑤⑥⑦⑧ as structure-based ones. Since `lrGAE` ②③④⑤ have implementations proposed in previous works Zhu et al. (2020); You et al. (2020); Kipf & Welling (2016a); Hou et al. (2022); Li et al. (2023b), we mainly provide the empirical results of `lrGAE` ⑥⑦⑧ in our experiments. The hyperparameters of all the baselines were configured according to the experimental settings officially reported by the authors and were then carefully tuned in our experiments to achieve their best results across all tasks. We also tune the hyperparameters of `lrGAE` variants for a fair comparison.

**Evaluation.** To provide a comprehensive benchmark, we conduct experiments on five graph learning tasks from node, link, subgraph, and graph levels, across homogeneous and heterogeneous graphs, i.e., node classification, link prediction, graph clustering, graph classification, and heterogeneous node classification.

- **Node classification (N).** Node classification is the most popular graph learning task, with the goal of assigning a class label to each node. In the graph self-supervised learning setting Li et al. (2023b), the GNN encoder is pretrained based on the pretext tasks to obtain the node embeddings. The final evaluation is done by fitting a linear classifier (i.e., a logistic regression model) on top of the frozen learned embeddings. We adopt the public splits for Cora, CiteSeer, and PubMed, and 8:1:1 training/validation/test splits for the remaining datasets. Classification *accuracy* is employed as the evaluation metric.

- **Link prediction (L).** For link prediction, the goal is to predict the existence of edges between pairs of nodes. For structure-based GAEs, we directly use the output of the structure decoder

---

[1] `https://github.com/pytorch/pytorch/blob/master/LICENSE`
[2] `https://github.com/pyg-team/pytorch_geometric/blob/master/LICENSE`

as the prediction, while for feature-based GAEs, we use the dot product of the learned embeddings of the node pairs as the prediction. We adopt the 85/5/10 training/validation/test splits for Cora, CiteSeer, and PubMed, as advocated by Kipf & Welling (2016a). The *Area Under the Curve (AUC)* and *Average Precision (AP)* are employed for evaluation.

- **Graph clustering (C).** Graph clustering involves grouping nodes in a graph such that nodes in the same group are more similar to each other compared to those in different groups. We perform K-means clustering on the learned embeddings to produce cluster assignments for each node and employ *Normalized Mutual Information (NMI)* as the evaluation metric.

- **Graph classification (G).** The graph classification task is similar to node classification but differs in that the class labels are assigned to the entire graph. Therefore, we perform `graph pooling` on the node embeddings to obtain the graph embedding. We follow the evaluation protocol of Hou et al. (2022; 2023), which uses a 10-fold cross-validation setting for train-test splits and reports the averaged results. Classification *accuracy* is employed as the evaluation metric.

- **Heterogeneous node classification (HN).** We also extend our experiments from common homogeneous graphs to heterogeneous ones, with heterogeneous node classification as the downstream task. In this task, we extend GAEs to heterogeneous graphs by reconstructing each graph view in a manner similar to homogeneous graphs and accumulating the loss from all views to learn the representations of each node type. We adopt the public train/valid/test splits as outlined in Lv et al. (2021) for all datasets. We use classification *accuracy* for the evaluation metric for the target node type in each dataset, i.e., *'author'* (DBLP), *'paper'* (ACM), and *'movie'* (IMDB).

For reproduction, we report the averaged results with standard standard deviations across 10 runs. All experiments are conducted on an NVIDIA RTX 3090 Ti GPU with 24 GB memory.

**Accessibility.** The source code for `lrGAE`, along with the baselines and scripts necessary to reproduce our experiments, is publicly available on GitHub [3]. This ensures that anyone with internet access can download and use the benchmark.

**Licensing.** The `lrGAE` benchmark is distributed under the MIT license, a permissive open-source license that allows users to freely use, modify, and distribute the code. This encourages widespread adoption and adaptation of the benchmark in various research and industrial applications.

**Tutorials and examples.** To assist new users, we provide tutorials and example notebooks that demonstrate how to use the benchmark for different tasks. These resources are designed to be beginner-friendly and help users understand the capabilities of the `lrGAE` framework.

**Community support.** We encourage collaboration and feedback from the community. Users can report issues, suggest improvements, and contribute to the project through the GitHub repository. Active community support ensures that the benchmark remains up-to-date and relevant.

# D  EXPERIMENTAL RESULTS

**Graph clustering.** Table 9 presents the graph clustering results on seven graph benchmarks. As we employ K-means as the downstream clustering method, we have observed high variances in the NMI results across the majority of datasets. In general, feature-based methods tend to demonstrate higher performance on NMI compared to structure-based methods. However, it is important to note that feature-based methods may suffer from higher memory overheads, particularly on datasets with high-dimensional input features. For structure-based GAEs, including `lrGAE` ⑥⑦⑧, they often achieve a more favorable trade-off between performance and scalability compared to feature-based methods. These structure-based approaches leverage the inherent binary graph topology to learn representations, which can be more efficient in terms of memory usage and computational complexity.

**Graph classification.** Table 10 presents the experimental results of graph classification on seven benchmarks. Note that GiGaMAE is not applicable for this specific task as it relies on node-level embedding methods, such as node2vec Grover & Leskovec (2016), to generate an initial embedding

---

[3] `https://anonymous.4open.science/r/lrGAE/`

Table 9: Graph clustering NMI (%) on seven benchmark datasets. In each column, the **boldfaced** score denotes the best result and the underlined score represents the second-best result. `OOM`: out-of-memory on an NVIDIA 3090ti GPU with 24GB memory.

| | Cora | CiteSeer | PubMed | Photo | Computers | CS | Physics |
|---|---|---|---|---|---|---|---|
| $GAE_f$ | $13.1_{\pm2.5}$ | $2.6_{\pm1.5}$ | $4.0_{\pm0.5}$ | $27.1_{\pm3.5}$ | $10.3_{\pm2.5}$ | $20.2_{\pm0.6}$ | OOM |
| GraphMAE | $54.3_{\pm2.1}$ | $44.9_{\pm1.7}$ | $33.2_{\pm0.4}$ | $66.9_{\pm3.2}$ | $56.8_{\pm2.4}$ | $76.2_{\pm0.8}$ | OOM |
| GraphMAE2 | $54.1_{\pm2.4}$ | $46.7_{\pm1.0}$ | $33.9_{\pm0.8}$ | $66.8_{\pm3.7}$ | $54.8_{\pm1.5}$ | $76.2_{\pm0.4}$ | OOM |
| AUG-MAE | $57.6_{\pm1.8}$ | $44.6_{\pm1.2}$ | $33.5_{\pm1.0}$ | $70.8_{\pm2.3}$ | $54.6_{\pm1.3}$ | OOM | OOM |
| GiGaMAE | $55.7_{\pm1.9}$ | $37.0_{\pm1.6}$ | $34.0_{\pm0.8}$ | $69.6_{\pm2.9}$ | $56.4_{\pm1.7}$ | $74.7_{\pm0.3}$ | OOM |
| GAE | $51.8_{\pm2.4}$ | $33.0_{\pm1.8}$ | $24.8_{\pm0.9}$ | $54.4_{\pm2.4}$ | $48.2_{\pm3.1}$ | $72.8_{\pm0.9}$ | $61.0_{\pm0.6}$ |
| S2GAE | $55.6_{\pm2.6}$ | $32.8_{\pm2.0}$ | $8.9_{\pm1.1}$ | $59.3_{\pm4.1}$ | $50.1_{\pm3.3}$ | $71.5_{\pm1.1}$ | $67.9_{\pm0.5}$ |
| MaskGAE | $58.0_{\pm2.4}$ | $43.8_{\pm1.4}$ | $28.6_{\pm1.2}$ | $58.2_{\pm3.5}$ | $56.3_{\pm2.9}$ | $76.8_{\pm0.8}$ | $74.9_{\pm0.9}$ |
| lrGAE ⑥ | $56.4_{\pm2.1}$ | $43.7_{\pm1.6}$ | $28.0_{\pm0.8}$ | $59.1_{\pm3.7}$ | $51.8_{\pm1.9}$ | $77.0_{\pm0.5}$ | $72.1_{\pm0.8}$ |
| lrGAE ⑦ | $59.0_{\pm2.6}$ | $44.9_{\pm1.8}$ | $27.3_{\pm0.9}$ | $64.4_{\pm3.8}$ | $57.0_{\pm2.2}$ | $77.2_{\pm0.7}$ | $66.0_{\pm0.4}$ |
| lrGAE ⑧ | $57.3_{\pm2.7}$ | $44.0_{\pm1.8}$ | $30.7_{\pm0.5}$ | $53.8_{\pm3.2}$ | $50.2_{\pm2.4}$ | $77.3_{\pm0.5}$ | $76.0_{\pm0.5}$ |

Table 10: Graph classification accuracy (%) on seven benchmark datasets. In each column, the **boldfaced** score denotes the best result and the underlined score represents the second-best result.

| | IMDB-B | IMDB-M | PROTEINS | COLLAB | MUTAG | REDDIT-B | NCI1 |
|---|---|---|---|---|---|---|---|
| $GAE_f$ | $74.4_{\pm0.8}$ | $52.5_{\pm0.7}$ | $75.3_{\pm1.2}$ | $76.9_{\pm1.5}$ | $87.0_{\pm1.2}$ | $72.8_{\pm2.5}$ | $74.5_{\pm1.3}$ |
| GraphMAE | $75.0_{\pm0.6}$ | $52.1_{\pm0.4}$ | $75.8_{\pm0.7}$ | $82.7_{\pm1.0}$ | $89.3_{\pm1.1}$ | $88.8_{\pm2.8}$ | $80.1_{\pm1.0}$ |
| GraphMAE2 | $75.5_{\pm0.7}$ | $52.7_{\pm0.6}$ | $75.4_{\pm0.5}$ | $81.7_{\pm0.8}$ | $89.5_{\pm1.5}$ | $88.6_{\pm2.3}$ | $82.3_{\pm0.9}$ |
| AUG-MAE | $75.6_{\pm1.1}$ | $52.2_{\pm1.0}$ | $73.5_{\pm0.8}$ | $79.9_{\pm0.4}$ | $89.8_{\pm1.3}$ | $88.3_{\pm3.1}$ | $78.8_{\pm1.2}$ |
| GiGaMAE | | | | N/A | | | |
| GAE | $75.1_{\pm0.7}$ | $51.5_{\pm1.5}$ | $76.6_{\pm0.9}$ | $80.1_{\pm0.6}$ | $89.5_{\pm1.5}$ | $82.5_{\pm2.5}$ | $73.8_{\pm1.7}$ |
| S2GAE | $73.6_{\pm0.6}$ | $52.5_{\pm0.9}$ | $76.0_{\pm0.3}$ | $82.2_{\pm0.4}$ | $85.7_{\pm0.8}$ | $89.4_{\pm2.7}$ | $77.2_{\pm0.8}$ |
| MaskGAE | $74.4_{\pm0.4}$ | $52.6_{\pm0.6}$ | $77.3_{\pm0.4}$ | $82.0_{\pm0.5}$ | $88.6_{\pm0.6}$ | $89.4_{\pm2.4}$ | $82.2_{\pm0.4}$ |
| lrGAE ⑥ | $74.0_{\pm0.3}$ | $52.2_{\pm0.4}$ | $77.2_{\pm0.6}$ | $82.3_{\pm0.7}$ | $89.9_{\pm0.4}$ | $88.5_{\pm1.8}$ | $81.5_{\pm0.6}$ |
| lrGAE ⑦ | $73.8_{\pm0.5}$ | $52.2_{\pm0.6}$ | $77.0_{\pm0.2}$ | $82.3_{\pm0.4}$ | $90.5_{\pm0.8}$ | $87.8_{\pm1.5}$ | $81.4_{\pm0.4}$ |
| lrGAE ⑧ | $73.8_{\pm0.3}$ | $52.7_{\pm0.5}$ | $76.1_{\pm0.4}$ | $82.2_{\pm0.2}$ | $89.3_{\pm0.7}$ | $88.4_{\pm1.0}$ | $81.2_{\pm0.7}$ |

for all nodes. As observed from Table 10, feature-based GAEs, particularly GraphMAE, achieve better results compared to structure-based GAEs. One possible reason for this disparity could be that reconstructing the structure is not an effective approach for graph-level tasks involving a set of disjoint graphs. In contrast, reconstructing node features can help avoid structural bias within batched graphs, leading to improved performance. By focusing on node feature reconstruction, feature-based GAEs can mitigate the challenges associated with capturing the structural information of disjoint graphs and provide more accurate representations for graph-level tasks.

**Heterogeneous node classification.** To further demonstrate the flexibility and comprehensibility of lrGAE across various graph learning scenarios, we benchmark experiments of GAEs on three heterogeneous graph datasets, including DBLP, ACM, and IMDB Lv et al. (2021); Li et al. (2023a). Table 11 presents the experimental results of different GAEs adapted to the heterogeneous node classification task. Several *supervised* heterogeneous GNNs, including HAN Wang et al. (2019), HGT Hu et al. (2020b), RGCN Schlichtkrull et al. (2018), and SHGN Lv et al. (2021) were listed Table 11 in for comparison. Note that we have excluded the comparison of feature-based GAEs due to missing attributes in several node types, which is unavailable to perform feature reconstruction for these datasets. As observed from Table 11, GAEs have shown comparable performance compared to heterogeneous GNNs even without the supervision of class labels. In particular, lrGAE ⑥⑦⑧ achieve generally better performance than other GAEs in most cases, on par with state-of-the-art heterogeneous GNNs.

Table 11: Heterogeneous node classification accuracy (%) on heterogeneous graphs benchmarks.

| | DBLP | ACM | IMDB |
|---|---|---|---|
| RGCN | $92.07_{\pm0.50}$ | $91.75_{\pm0.35}$ | $65.21_{\pm0.73}$ |
| HAN | $92.05_{\pm0.62}$ | $90.79_{\pm0.43}$ | $64.63_{\pm0.58}$ |
| HGT | $93.49_{\pm0.25}$ | $91.15_{\pm0.71}$ | $67.20_{\pm0.57}$ |
| SHGN | $94.20_{\pm0.31}$ | $93.35_{\pm0.45}$ | $\mathbf{67.36}_{\pm0.57}$ |
| $\text{GAE}_f$ | | $\text{N/A}^\dagger$ | |
| GraphMAE | | $\text{N/A}^\dagger$ | |
| GraphMAE2 | | $\text{N/A}^\dagger$ | |
| Aug-MAE | | $\text{N/A}^\dagger$ | |
| GiGaMAE | | $\text{N/A}^\dagger$ | |
| GAE | $94.1_{\pm0.3}$ | $93.3_{\pm0.5}$ | $65.4_{\pm0.2}$ |
| S2GAE | $\mathbf{94.8}_{\pm0.4}$ | $93.8_{\pm0.7}$ | $65.7_{\pm0.5}$ |
| MaskGAE | $94.2_{\pm0.2}$ | $93.7_{\pm0.2}$ | $65.9_{\pm0.7}$ |
| lrGAE ⑥ | $94.4_{\pm0.2}$ | $93.5_{\pm0.4}$ | $66.0_{\pm0.5}$ |
| lrGAE ⑦ | $94.6_{\pm0.3}$ | $94.0_{\pm0.3}$ | $65.2_{\pm0.6}$ |
| lrGAE ⑧ | $94.3_{\pm0.2}$ | $\mathbf{94.1}_{\pm0.1}$ | $66.3_{\pm0.4}$ |

$^\dagger$ Not applicable due to missing node attributes.

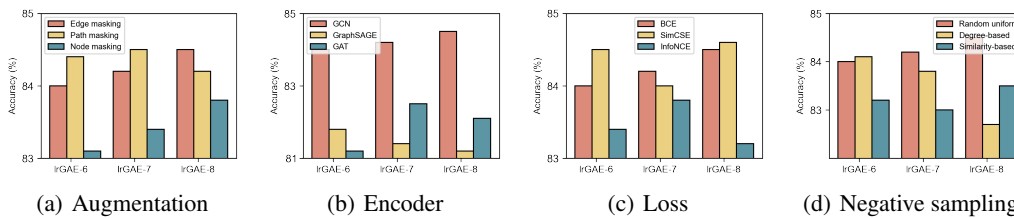

|     (a) Augmentation     |     (b) Encoder     |     (c) Loss     |     (d) Negative sampling     |

Figure 3: Ablation on three `lrGAE` variants with different augmentation (masking) strategies, encoder networks, contrastive loss, and negative sampling strategies.

# E   ABLATION STUDIES

In this section, we perform ablation studies on the key components of `lrGAE`, i.e., augmentation, encoders, contrastive loss, and negative sampling strategies. We opt for `lrGAE` ⑥⑦⑧, three advanced GAEs that incorporate different contrastive schemes as the comparison methods. We conduct experiments on node classification tasks using the Cora dataset. The experimental results were averaged over 10 runs.

**Augmentations.** We first conduct ablation studies on the augmentation techniques, which include edge masking Rong et al. (2020), path masking Li et al. (2023b), and node masking You et al. (2020). Figure 3(a) presents the ablation results of `lrGAE` ⑥⑦⑧ using various masking strategies. As observed, the performance of `lrGAE` ⑥⑦⑧ varies significantly with different masking strategies. Basically, edge masking and path masking are better choices than node masking. The observation meets our intuition. This highlights the effectiveness of edge and path masking techniques in enhancing model performance, likely due to their ability to capture more nuanced relationships within the data compared to node masking.

**Encoder networks.** Given that most of the decoder networks in GAEs are MLPs, we will focus the ablation experiments only on the encoder networks. The encoder plays a crucial role in mapping graphs into low-dimensional representations. To investigate the potential of designing effective GAEs, we perform ablation studies on Cora using three different GNN encoders, including GCN Kipf & Welling (2016b), GraphSAGE Hamilton et al. (2017), and GAT Veličković et al. (2018). The results, shown in Figure 3(b), demonstrate that GCN is the most effective encoder architecture across all three `lrGAE` variants. `lrGAE` ⑥⑦⑧ with GCN as the encoder consistently outperforms GAT and GraphSAGE in all cases by large margins. This observation is consistent with the conclusions of

previous studies Li et al. (2023b); Velickovic et al. (2019); Thakoor et al. (2021); Zhu et al. (2021); Zhang et al. (2021).

**Contrastive loss.** Next, we conduct ablation studies on the contrastive loss in GAEs, using binary cross-entropy (BCE), SimCSE Gao et al. (2021), and InfoNCE van den Oord et al. (2018). SimCSE and InfoNCE are two effective losses utilized in contrastive learning literature. SimCSE focuses on maximizing the similarity between augmented views of the same instance, whereas InfoNCE involves distinguishing positive samples from a set of negative samples. We present the results in Figure 3(c). The results reveal that although BCE is the most widely used loss function in traditional GAEs, SimCSE shows potential as a better alternative, outperforming BCE and InfoNCE in 2 out of 3 `lrGAE` variants. This finding points to SimCSE as a promising direction for future research and optimization in contrastive learning within GAEs.

**Negative samples.** We explore different negative sampling strategies for `lrGAE`, which are detailed below:

- **Random uniform sampling.** Negative edges are sampled uniformly from the set of all possible non-existent edges. This is the simplest approach but may not always yield the best results.
- **Degree-based sampling.** Negative edges are sampled based on the degree of nodes. The core idea is that nodes with higher degrees are more likely to be involved in many connections, making them more likely to be sampled as negative examples.
- **Similarity-based sampling.** Negative edges are sampled based on the similarity or distance between nodes in the graph. For instance, edges between nodes that are far apart are more likely to be sampled as negative. In this case, we use cosine similarity as the distance measure.

The ablation results are shown in Figure 3(d). We observe that random negative sampling generally outperforms other strategies in most cases. Additionally, degree-based sampling shows promise as an effective negative sampling strategy. While similarity-based sampling was expected to improve performance by distinguishing nodes that are more dissimilar, it did not perform as well as the random or degree-based approaches. The possible reason could be that GAEs necessitate more *hard* negative samples, whereas similarity-based sampling might only furnish *easy* ones.

**Remarks.** We leave two additional remarks for our ablation studies. Firstly, the ablation experiments were only conducted on the node classification task and one dataset, which may limit the generalizability of the findings to other tasks and datasets. Secondly, we adopted three new `lrGAE` variants as examples, which provide a representative but not an exhaustive exploration of the possible configurations of GAEs. Further studies with more tasks, datasets, and variants would be beneficial to fully assess the robustness of the negative sampling strategies and the performance of `lrGAE`.

Table 12: Node classification on large scale graphs ogbn-arXiv, ogbn-MAG, ogbn-Products Hu et al. (2020a).

| | arXiv | MAG | Products |
|---|---|---|---|
| $GAE_f$ | $64.2_{\pm 0.7}$ | $25.7_{\pm 0.4}$ | $73.5_{\pm 0.6}$ |
| GraphMAE | $71.0_{\pm 0.4}$ | $32.2_{\pm 0.3}$ | $78.9_{\pm 0.4}$ |
| GraphMAE2 | $71.6_{\pm 0.2}$ | $32.7_{\pm 0.2}$ | $81.0_{\pm 0.2}$ |
| AUG-MAE | OOM | OOM | OOM |
| GiGaMAE | OOM | OOM | OOM |
| GAE | $68.9_{\pm 0.3}$ | $30.1_{\pm 0.3}$ | $77.8_{\pm 0.3}$ |
| S2GAE | $70.5_{\pm 0.2}$ | $31.7_{\pm 0.1}$ | $80.0_{\pm 0.5}$ |
| MaskGAE | $71.2_{\pm 0.1}$ | $32.8_{\pm 0.2}$ | $79.6_{\pm 0.4}$ |
| `lrGAE` ⑥ | $70.8_{\pm 0.2}$ | $32.7_{\pm 0.1}$ | $80.7_{\pm 0.1}$ |
| `lrGAE` ⑦ | $71.4_{\pm 0.1}$ | $32.9_{\pm 0.1}$ | $81.9_{\pm 0.3}$ |
| `lrGAE` ⑧ | $71.9_{\pm 0.1}$ | $32.2_{\pm 0.1}$ | $82.3_{\pm 0.2}$ |

