# OpenReview forum: "Revisiting and Benchmarking Graph Autoencoders: A Contrastive Learning Perspective"
_ICLR.cc/2025/Conference — ICLR 2025 Conference Withdrawn Submission_

### Official Review · Reviewer_x6w6 · 2024-10-31

**Soundness:** 3
**Presentation:** 3
**Contribution:** 3
**Rating:** 6
**Confidence:** 5

**Summary:**

This paper presents lrGAE (left-right GAE), i.e., a comprehensive benchmark for graph autoencoders (GAEs) in self-supervised learning (SSL). It aims to conceptually and methodologically bridge the gap between GAEs and contrastive learning. The proposed lrGAE enables an extensive comparison of GAEs across eight unique configurations on seventeen real-world datasets, covering a range of scenarios.

**Strengths:**

1. This paper formalizes a very critical and interesting problem, i.e., exploring the connection between generative and contrastive learning in graph SSL. It provides insight into their relationship through the lens of the loss function and breaks down GAEs based on the main components of contrastive learning.

2. The paper is well-structured and easy to understand, presenting an evaluation across a diverse set of datasets and task types.

3. lrGAE is available as an open-source tool, which can help researchers test their model and keep track of the current research frontier.

**Weaknesses:**

1. Lack of relevant GAEs baselines/references - Current discussion of generative SSL is mainly contained within masked GAEs, while an important research direction involving spectral GAEs, such as asymmetric spectral filters in GALA [1], low-pass filters in GATE [2, 3], and augmentation-adaptive filters in WGDN [4], is missing. Clarification on how spectral GAEs fit into contrastive-based categories is needed.

2. Inadequate detail on experimental settings and analysis explanations.

    2.1 Critical information on experimental configurations, such as hidden dimensions, training epochs, learning rate ranges, scheduling strategies, and hyperparameter optimization techniques, is not provided. These details are essential for the comprehensiveness and credibility of a benchmark paper.

   2.2 Some baseline methods experience out-of-memory issues for the Physics dataset in Tables 4 and 9. While the authors attribute this to large feature dimensions, reducing the hidden dimension, as suggested by [4], could be a solution. It is advisable to explicitly state and adjust hidden dimensions to facilitate a more balanced and thorough comparison.

   2.3 The authors argue that structure-based GAEs underperform compared to feature-based GAEs due to structural bias. However, for graph-level tasks, attributes are often hand-crafted structural features (e.g., degrees). A more plausible explanation might be that feature-based GAEs extract more nuanced information through the interplay between graph structures and derived attributes.

3. It's recommended to include more experimental results in the main text, as expected of a benchmark paper. Currently, extensive results are deferred to the appendix to adhere to the 9-page limit, but the latest guidelines allow up to 10 pages.

[1] Jiwoong Park, et al. Symmetric Graph Convolutional Autoencoder for Unsupervised Graph Representation Learning. In ICCV 2019.

[2] Amin Salehi and Hasan Davulcu, Graph Attention AutoEncoders. In ICTAI 2020.

[3] Muhammet Balcilar, et al. Analyzing the expressive power of graph neural networks in a spectral perspective. In ICLR 2021.

[4] Jiashun Cheng, et al. Wiener Graph Deconvolutional Network Improves Graph Self-Supervised
Learning. In AAAI 2023.

**Questions:**

See Weakness.

---

> ### Author Response · Authors · 2024-11-19
>
> We appreciate the reviewer for detailed comments and suggestive feedback. We want to clarify some misunderstandings that caused some of your concerns.
>
> > W1: Lack of relevant GAEs baselines/references - Current discussion of generative SSL is mainly contained within masked GAEs, while an important research direction involving spectral GAEs, such as asymmetric spectral filters in GALA [1], low-pass filters in GATE [2, 3], and augmentation-adaptive filters in WGDN [4], is missing. Clarification on how spectral GAEs fit into contrastive-based categories is needed.
>
> Thank you for pointing out these important works. We apologize for not considering them, as our primary focus was on more advanced GAEs, specifically masked graph autoencoders. We will expand the context of the lrGAE framework to include additional GAE implementations in the next revision. We appreciate your feedback and will ensure that these works are properly integrated into the discussion to provide a more comprehensive comparison.
>
> > W2: Inadequate detail on experimental settings and analysis explanations.
>
> Thank you for the insightful comments. For the experimental configurations, we have provided detailed scripts in the code repository, which we believe are sufficient for readers to reproduce all the experiments. The out-of-memory issues with the Physics dataset were caused by the large dimensions of the node features (i.e., 8k+), which are prohibitive for feature reconstruction. We followed the experimental setting in GraphMAE and selected the best hidden dimension (i.e., 1024) for reproducibility. We will take all your suggestions into account and rerun the experiments with a smaller hidden dimension.
>
> > W3: It's recommended to include more experimental results in the main text, as expected of a benchmark paper. Currently, extensive results are deferred to the appendix to adhere to the 9-page limit, but the latest guidelines allow up to 10 pages.
>
> Thank you for your suggestion! According to the ICLR 2025 submission guidelines at https://iclr.cc/Conferences/2025/CallForPapers, it is recommended to use 9 pages for the main text:
>
>  "We encourage authors to be crisp in their writing by submitting papers with 9 pages of main text. We recommend that authors only use the longer page limit in order to include larger and more detailed figures. However, authors are free to use the pages as they wish, as long as they obey the page limits."
>
> We initially submitted our manuscript with 9 pages of main text to align with these guidelines. We will now re-structure our paper to include additional experimental results in the main text.
>
> ---
>
> We hope these responses help to alleviate your concerns, and look forward to receiving feedback regarding any remaining questions at your earliest convenience.

---

### Official Review · Reviewer_bS1D · 2024-11-02

**Soundness:** 2
**Presentation:** 3
**Contribution:** 2
**Rating:** 3
**Confidence:** 4

**Summary:**

This paper re-evaluates Graph Auto-Encoders (GAEs) through the lens of Graph Contrastive Learning (GCL), uncovering the conceptual and methodological links between the two. Furthermore, the authors present a unified framework, lrGAE, and evaluate the performance of various GAEs on different downstream tasks to validate the properties of the proposed methods.

**Strengths:**

1.	The writing is clear, and the overall structure is easy to understand.
2.	The paper provides a systematic and theoretical analysis of the connection between feature-based GAEs and graph contrastive learning (GCL).

**Weaknesses:**

1.	The innovation appears limited, as most insights are derived from previous works, including the equivalence between structure-based GAEs and GCLs. The key viewpoint about feature-based methods, two rooted subgraphs are considered positive pairs, is also inspired by another similar work.
2.	The primary contribution of this paper seems to be a new perspective on current graph self-supervised learning (SSL) methods. All variants of lrGAE achieve performance levels similar to existing models, with no improvements in link prediction or node classification tasks. It would be more persuasive if the effectiveness of this new insight were demonstrated through performance enhancement or mitigation of specific problems.
3.	The paper lacks a novel, comprehensive theoretical analysis of the link between GAE and GCL. It relies heavily on theoretical conclusions from other works, such as [1], without thoroughly examining the equivalence between these two unsupervised methods. According to lines 237-247 and 260-261, the training process for structure-based and feature-based GAEs is only considered an alignment-loss minimization procedure, while the uniformity loss (Eq 7) remains unexplored. The statement "GAEs could be regarded as an approximated contrastive learning" is not convincing as a conclusion about the connection between GAE and GCL.
4.	A good benchmark should introduce new metrics/datasets, focusing on specific domain issues or commonly overlooked problems within the field. Relying on common metrics such as "Link Prediction" and "Node Classification," along with merely combining different contrastive views, The current contribution may not produce a proper benchmark.

**Questions:**

1.	Lack a KL term in Eq5 ?
Please see the above Weaknesses.

---

> ### Author Response · Authors · 2024-11-19
>
> We thank the reviewers for reading our paper and providing detailed review on our submission. We respond to the reviewers’ concerns and questions one by one.
>
> > W1: The innovation appears limited, as most insights are derived from previous works, including the equivalence between structure-based GAEs and GCLs. The key viewpoint about feature-based methods, two rooted subgraphs are considered positive pairs, is also inspired by another similar work.
>
> Our work differs from previous studies that focus on a single aspect of GAEs, such as graph structure or node features. Specifically, our work extends the scope of prior research by providing a deeper analysis of the connections between GAEs and GCL. Previous works (e.g., MaskGAE) primarily focus on structure-based GAEs, without offering an in-depth analysis of feature-based GAEs or presenting additional theoretical results beyond the context of MaskGAE. In contrast, our work broadens the scope by considering both graph structure and node features, providing a more comprehensive analysis of GAEs in the context of GCL.
>
> > W2: The primary contribution of this paper seems to be a new perspective on current graph self-supervised learning (SSL) methods. All variants of lrGAE achieve performance levels similar to existing models, with no improvements in link prediction or node classification tasks. It would be more persuasive if the effectiveness of this new insight were demonstrated through performance enhancement or mitigation of specific problems.
>
> lrGAE-6, lrGAE-7, and lrGAE-8 are three examples of GAEs designed with a contrastive learning perspective, particularly in terms of their contrastive views. Indeed, more GAEs can be developed by exploring augmentation strategies, encoder/decoder architectures, contrastive loss functions, and even negative example selection. The proposed lrGAE-6, -7, and -8 demonstrate the potential of leveraging contrastive learning. While they are not intended as new state-of-the-art methods in graph self-supervised learning, they nonetheless achieve superior performance in several scenarios.
>
> > W3: According to lines 237-247 and 260-261, the training process for structure-based and feature-based GAEs is only considered an alignment-loss minimization procedure, while the uniformity loss (Eq 7) remains unexplored. The statement "GAEs could be regarded as an approximated contrastive learning" is not convincing as a conclusion about the connection between GAE and GCL.
>
> In structure-based GAEs, the uniformity loss is implemented (in an approximate manner) via the non-edge part of the loss function. In feature-based GAEs, it is true that the uniformity regularization is not present in the loss, which may lead to degenerate solutions. Masked GAE alleviates this degeneracy, as stated in [1]. Introducing a formal uniformity regularization term into our framework may further improve model performance, which we leave for future exploration.
>
> > W4: A good benchmark should introduce new metrics/datasets, focusing on specific domain issues or commonly overlooked problems within the field. Relying on common metrics such as "Link Prediction" and "Node Classification," along with merely combining different contrastive views, The current contribution may not produce a proper benchmark.
>
> We respectfully disagree with your point that a good benchmark should introduce new metrics/datasets. We believe that **a good benchmark does not need to rely on new metrics or datasets** to be impactful. Instead, its value lies in its ability to rigorously evaluate models within a well-defined scope, provide actionable insights, and inspire new directions in research.
>
> Rather than introducing new metrics or datasets, our benchmark offers a **new perspective by connecting GAEs with GCL**, providing deeper insights into their relationships and fostering novel research directions in this area. This approach goes beyond a simple collection of existing works by highlighting previously unexplored aspects, which we believe is a significant contribution.
>
> > Q1: Lack a KL term in Eq5 ?
>
> Equation 5 denotes the feature-based GAEs that adopt feature reconstruction for self-supervised learning. In this regard, we don't believe there is a missing KL term.
>
> ---
>
> We would appreciate it if you could let us know if our response has addressed your concerns about our work. If not, please kindly let us know what we are lacking so that we can provide better clarification. Thank you for taking the time to review our work.
>
> [1] How Mask Matters: Towards Theoretical Understandings of Masked Autoencoders. NeurIPS 2022

---

> > ### Comment · Reviewer_bS1D · 2024-11-24
> >
> > Thank you for the response. Unfortunately, the response does not adequately address my concerns regarding the novelty, the contribution to the community, and the demonstration of effectiveness. Therefore, I maintain my score.

---

### Official Review · Reviewer_QPeH · 2024-11-03

**Soundness:** 3
**Presentation:** 3
**Contribution:** 2
**Rating:** 5
**Confidence:** 4

**Summary:**

This paper focuses on self-supervised representation learning on graphs (especially node-level graph autoencoders). The paper proposes that recent graph mask autoencoder methods can be understood from a contrastive learning perspective. Based on this perspective, the paper presents a design space for graph autoencoders, categorizes current methods, and proposes several novel methods based on this design space. In summary, this paper provides a new perspective for understanding graph autoencoders.

**Strengths:**

1. This paper provides a unified perspective for understanding graph contrastive learning and various autoencoder methods, which is very meaningful for further understanding graph self-supervised learning.

2. The paper's writing quality and presentation quality are high.

3. The authors provide well-documented code and documentation for reproduction.

**Weaknesses:**

1. Although there hasn't been a systematic study of (masked) graph autoencoders, this paper isn't the first to propose connecting graph autoencoders with contrastive learning. For example, in [1,2], it was already proposed that adjacent node pairs could be used as positive pairs in contrastive learning. Therefore, in my view, the idea in this paper isn't novel (though I appreciate this paper's contribution as a systematic overview of graph autoencoders).

2. The paper focuses mainly on summarizing previous methods. Although Table 1 proposes three new methods (lrGAE6,7,8), specific details about these methods and their implementations cannot be found in the main text.

3. The proposed methods didn't achieve particularly impressive results.

References:

[1]  Lee, N., Lee, J., and Park, C. Augmentation-free self-supervised learning on graphs. arXiv preprint arXiv:2112.02472, 2021

[2]  Zhang, Hengrui, et al. "Localized contrastive learning on graphs." arXiv preprint arXiv:2212.04604, 2022.

**Questions:**

1. Besides conceptual and perspective-based contributions, does LrGAE have any empirical contributions compared to current methods?

2. Do the node pairs in Table 1 refer to the objects of reconstruction? For example, when v = u, does it refer to feature reconstruction, and when v ≠ u, does it refer to edge reconstruction?

---

> ### Author Response · Authors · 2024-11-19
>
> We thank the reviewers for reading our paper and providing detailed review on our submission. We respond to the reviewers’ major concerns and questions below.
>
> > W1: Although there hasn't been a systematic study of (masked) graph autoencoders, this paper isn't the first to propose connecting graph autoencoders with contrastive learning. For example, in [1,2], it was already proposed that adjacent node pairs could be used as positive pairs in contrastive learning. Therefore, in my view, the idea in this paper isn't novel (though I appreciate this paper's contribution as a systematic overview of graph autoencoders).
>
> Our work extends the scope of previous studies by providing a deeper analysis of the connections between GAEs and GCL. Existing works that use adjacent node pairs as positive pairs in contrastive learning do not fully uncover the key connections between GAEs and GCL. Instead, they focus only on node neighbors as positive pairs without exploring the underlying successes of GAEs or GCL. We not only bridge the gap in understanding the hidden connections between GAEs (both structure-based and feature-based) and GCL, but also adapt current GAEs to our framework, re-implementing them from the perspective of graph contrastive learning. This, we believe, will take the research on GAEs to a whole new perspective.
>
> > W2: The paper focuses mainly on summarizing previous methods. Although Table 1 proposes three new methods (lrGAE6,7,8), specific details about these methods and their implementations cannot be found in the main text.
>
> Our work is not a simple implementation or collection of existing GAE works.  Instead, we adapt current GAEs to our framework and re-implement them from the perspective of graph contrastive learning. This, we believe, will take the research on GAEs to a whole new perspective. In addition, the proposed lrGAE-6,7,8 are showcasing the possibility of unleashing the power of contrastive learning, rather than proposed as new state-of-the-arts in graph self-supervised learning, they achieve superior performance in several cases though.
>
> > W3: The proposed methods didn't achieve particularly impressive results.
>
> lrGAE-6, lrGAE-7, and lrGAE-8 are three examples of GAEs designed with a contrastive learning perspective, particularly in terms of their contrastive views. Indeed, more GAEs can be developed by exploring augmentation strategies, encoder/decoder architectures, contrastive loss functions, and even negative example selection. The proposed lrGAE-6, -7, and -8 demonstrate the potential of leveraging contrastive learning. While they are not intended as new state-of-the-art methods in graph self-supervised learning, they nonetheless achieve superior performance in several scenarios.
>
> ---
>
> Thank you again for taking the time to review our paper. We hope our responses could clarify your concerns, and hope you will consider increasing your score. If we have left any notable points of concern unaddressed, please do share and we will attend to these points.

---

> > ### Comment · Reviewer_QPeH · 2024-11-26
> > **Response to the author**
> >
> > Thank you for the author's response. Unfortunately, your response did not address my concerns about this paper. For example, regarding the second question, the author did not answer the specific implementation details of lrGAE6,7,8. After reading other reviewers' comments, I believe that the current quality of this paper is not sufficient to warrant its publication at ICLR. Therefore, I maintain my score.

---

### Official Review · Reviewer_3g52 · 2024-11-03

**Soundness:** 3
**Presentation:** 2
**Contribution:** 1
**Rating:** 5
**Confidence:** 4

**Summary:**

This paper presents a novel framework, lrGAE (left-right Graph Autoencoder), which combines principles from contrastive learning with graph autoencoders (GAEs) to enhance representation learning for graph-structured data.

**Strengths:**

A comprehensive analysis of self-supervised learning on graphs, especially understanding graph autoencoder from the perspective of graph contrastive learning.

**Weaknesses:**

1. In [1], GAEs has already been connected to GCLs. While the paper introduce a more comprehensive analysis, it is a summary or deep understanding.

2. The core components of lrGAE, including augmentation strategies, contrastive views, encoder-decoder networks, and the use of contrastive loss without negative samples, are extensively covered in prior literature.

3. From the experimental results, the proposed lrGAE (6,7,8) cannot achieve better performance compared to masked graph autoencoder model MaskGAE.

[1] Li, J., Wu, R., Sun, W., Chen, L., Tian, S., Zhu, L., Meng, C., Zheng, Z. and Wang, W., 2023, August. What's Behind the Mask: Understanding Masked Graph Modeling for Graph Autoencoders. KDD

**Questions:**

What is the performance of the proposed method on large-scale datasets?

---

> ### Author Response · Authors · 2024-11-19
>
> We thank the reviewers for reading our paper and providing detailed review on our submission. We respond to the reviewers’ concerns and questions one by one.
>
> > W1: In [1], GAEs has already been connected to GCLs. While the paper introduce a more comprehensive analysis, it is a summary or deep understanding.
>
> Our work extends the scope of previous studies by providing a deeper analysis of the connections between GAEs and GCL. Unlike prior works (e.g., MaskGAE), which primarily focus on structure-based GAEs, our work offers a more in-depth analysis of feature-based GAEs and presents additional theoretical results beyond the context of MaskGAE. By doing so, we offer a unified framework that integrates theoretical insights and practical implementations for structure-based and feature-based GAEs, which paves the way for future research in graph self-supervised learning.
>
> > W2: The core components of lrGAE, including augmentation strategies, contrastive views, encoder-decoder networks, and the use of contrastive loss without negative samples, are extensively covered in prior literature.
>
> We believe there may have been some misunderstanding about our contributions. Please note that our goal is not to propose several new components within the context of GCL. Instead, we aim to adapt GAEs to existing GCL frameworks, which share the same paradigms and components, such as augmentation strategies and contrastive views. This adaptation allows us to unify GAEs and GCL under a common framework and highlights their shared principles and demonstrating how GAEs can benefit from advancements in GCL.
>
> > W3: From the experimental results, the proposed lrGAE (6,7,8) cannot achieve better performance compared to masked graph autoencoder model MaskGAE.
>
> lrGAE-6, lrGAE-7, and lrGAE-8 are three examples of GAEs designed with a contrastive learning perspective, particularly in terms of their contrastive views. Indeed, more GAEs can be developed by exploring augmentation strategies, encoder/decoder architectures, contrastive loss functions, and even negative example selection. The proposed lrGAE-6, -7, and -8 demonstrate the potential of leveraging contrastive learning. While they are not intended as new state-of-the-art methods in graph self-supervised learning, they nonetheless achieve superior performance in several scenarios.
>
> > Q1: What is the performance of the proposed method on large-scale datasets?
>
> Thank you for your question. The scalability of both structure-based GAEs and feature-based GAEs has been demonstrated in the literature [1,2]. Most implemented GAEs, including the lrGAE variants, can scale to significantly larger graphs without the need for sophisticated engineering tricks. Empirically, we provide experimental results on three large-scale datasets—arXiv, MAG, and Products from the OGB benchmark—to validate this scalability.
>
>
> |               | **arXiv**           | **MAG**             | **Products**        |
> | ------------- | ------------------- | ------------------- | ------------------- |
> | **GAE$_f$**   | 64.2$_{\pm0.7}$     | 25.7$_{\pm0.4}$     | 73.5$_{\pm0.6}$     |
> | **GraphMAE**  | 71.0$_{\pm0.4}$     | 32.2$_{\pm0.3}$     | 78.9$_{\pm0.4}$     |
> | **GraphMAE2** | 71.6$_{\pm0.2}$     | 32.7$_{\pm0.2}$     | 81.0$_{\pm0.2}$     |
> | **AUG-MAE**   | OOM                 | OOM                 | OOM                 |
> | **GiGaMAE**   | OOM                 | OOM                 | OOM                 |
> | **GAE**       | 68.9$_{\pm0.3}$     | 30.1$_{\pm0.3}$     | 77.8$_{\pm0.3}$     |
> | **S2GAE**     | 70.5$_{\pm0.2}$     | 31.7$_{\pm0.1}$     | 80.0$_{\pm0.5}$     |
> | **MaskGAE**   | 71.2$_{\pm0.1}$     | 32.8$_{\pm0.2}$     | 79.6$_{\pm0.4}$     |
> | **lrGAE-6**   | 70.8$_{\pm0.2}$     | 32.7$_{\pm0.1}$     | 80.7$_{\pm0.1}$     |
> | **lrGAE-7**   | 71.4$_{\pm0.1}$     | **32.9$_{\pm0.1}$** | 81.9$_{\pm0.3}$     |
> | **lrGAE-8**   | **71.9$_{\pm0.1}$** | 32.2$_{\pm0.1}$     | **82.3$_{\pm0.2}$** |
>
> Reference
>
> [1] Li, Jintang, et al. "What's Behind the Mask: Understanding Masked Graph Modeling for Graph Autoencoders." Proceedings of the 29th ACM SIGKDD Conference on Knowledge Discovery and Data Mining. 2023.
>
> [2] Hou, Zhenyu, et al. "Graphmae2: A decoding-enhanced masked self-supervised graph learner." *Proceedings of the ACM web conference 2023*. 2023.

---

### Official Review · Reviewer_p32P · 2024-11-04

**Soundness:** 3
**Presentation:** 3
**Contribution:** 2
**Rating:** 3
**Confidence:** 4

**Summary:**

The paper analyzes the connection between two graph self-supervised learning paradigms, i.e., graph contrastive learning and graph autoencoders. Then, it enhances graph autoencoders with contrastive learning steps, proposing a new model. Finally, it conducts experiments on various graph SSL models on different tasks, including node classification, link prediction, and clustering.

**Strengths:**

1. The paper provides a comprehensive analysis on different types of graph CL and graph AE methods, identifying the design recipe of graph AE models. Table 1 is good.
2. The experiments are comprehensive, from my perspective.

**Weaknesses:**

1. Originality: I have doubts about the originality of the work. The abstract states that "the underlying mechanisms of GAEs are not well understood, and a comprehensive benchmark for GAEs is still lacking." However, I believe this is not accurate. For instance, S2GAE already analyzed the connection between graph masked autoencoders and contrastive learning. It is well recognized that "GAEs, whether employing structure or feature reconstruction, with or without masked autoencoding, implicitly perform graph contrastive learning on two paired subgraph views." Moreover, I did not find that this paper proposes a more comprehensive benchmark, since most of the baselines are already covered in recent papers. Contrastive learning, MAE, and graph learning have been long-standing subjects of study. While reading this paper, I felt that many claims were not very novel, suggesting that this work is rather incremental.

2. Clarity: I believe the authors attempted to provide a detailed explanation of the proposed idea in the methodology part, but the writing could be more concise. The narrative is quite extended, and I was lost as to what the KEY contribution of this work is (rather than several marginal contributions put together). Additionally, there seems to be a typo in the explanation of Eq 8.

**Questions:**

Can you summarize what is THE KEY contribution of this paper?

---

> ### Author Response · Authors · 2024-11-19
>
> We appreciate the reviewer for detailed comments and suggestive feedback. We want to clarify some misunderstandings that caused some of your concerns.
>
> > W1: Originality: I have doubts about the originality of the work. The abstract states that "the underlying mechanisms of GAEs are not well understood, and a comprehensive benchmark for GAEs is still lacking." However, I believe this is not accurate. For instance, S2GAE already analyzed the connection between graph masked autoencoders and contrastive learning. It is well recognized that "GAEs, whether employing structure or feature reconstruction, with or without masked autoencoding, implicitly perform graph contrastive learning on two paired subgraph views." Moreover, I did not find that this paper proposes a more comprehensive benchmark, since most of the baselines are already covered in recent papers. Contrastive learning, MAE, and graph learning have been long-standing subjects of study. While reading this paper, I felt that many claims were not very novel, suggesting that this work is rather incremental.
>
> Firstly, our work extends the scope of previous studies by providing a deeper analysis of the connections between GAEs and GCL. Unlike prior works (e.g., MaskGAE), which primarily focus on structure-based GAEs, our work offers a more in-depth exploration of feature-based GAEs.
>
> Secondly, existing works (e.g., S2GAE) only scratch the surface of the deep connections between feature-based GAEs and GCL, lacking theoretical analysis. Our work addresses this gap and offers a novel perspective on feature-based GAEs.
>
> Thirdly, our work is not merely an implementation or collection of existing GAE methods. Instead, we adapt current GAEs to our framework and re-implement them through the lens of graph contrastive learning. We believe this approach will advance the research on GAEs by introducing a fresh and more comprehensive perspective.
>
> > W2: The narrative is quite extended, and I was lost as to what the KEY contribution of this work is (rather than several marginal contributions put together).
>
> Thank you for the feedback. We appreciate your comments and will work to clarify the narrative. The key contribution of our work lies in providing a unified framework that bridges the gap between GAEs and GCL. We summarize the key contributions of our work below:
>
> - **A first-ever GAE benchmark**: Our work is the first GAE benchmark that provides a comprehensive recipe for GAEs.
> - **A unified GAE framework with a new perspective from GCL**: Rather than being a simple implementation or collection of existing GAE methods, our work delves deeper into the connections between GAEs and GCL, offering insights that could inspire further research in this area.
> - **Comprehensive experiments**: We conduct extensive benchmarking of GAEs across a wide range of graph learning tasks and graph modalities.
>
> These contributions aim to advance the understanding and applications of GAEs in a cohesive and impactful manner. We will refine the presentation in our revision to ensure the central contributions are more prominently highlighted.
>
> ---
>
> If there are any remaining concerns about the novelty of our work, please let us know. We are more than happy to provide additional information to help clarify the situation.

---

> > ### Comment · Reviewer_p32P · 2024-11-24
> >
> > 1. I would avoid saying "the underlying mechanisms of GAEs are not well understood" in a domain where many studies have been done. This is also obvious in other reviewers' comments that people feel negative about this type of claim. You see that every reviewer can find some related work.
> > 2. I would suggest the authors to be more **specific** in your abstract + introduction about what specific novel contributions/advances/discoveries/methods you have proposed. Putting more math or making the paper hard to read does not mean you go deeper. Before you go deeper, please be specific where you want to going deeper and why. "Our work addresses this gap and offers a novel perspective on feature-based GAEs" --- this is good, much better than the tons of ambitiou claims in our original version.
> > 3. "A first-ever GAE benchmark" or "A unified GAE framework with a new perspective from GCL" are not exciting to me. Having a benchmark is not importnat for GAE-related studies. This topic has been studied for several years, and most datasets and baselines are open-sourced.

---

### Author Response · Authors · 2024-11-24
**Gentle Reminder to Reviewers**

Dear Reviewers,

We sincerely appreciate your insightful review and feedback comments. All comments have greatly enhanced our paper.

**As the author-reviewer discussion deadline is approaching, we would like to check if you have any other remaining concerns about our paper.** We have faithfully responded to all your comments and revise our paper according to your suggestions. If our responses have adequately addressed your concerns, we kindly hope that you can consider increasing the score. We understand that you have a demanding schedule, and we appreciate the time and effort you dedicate to reviewing our paper.

Kind regards,

Authors

---

### Note · Authors · 2024-12-03

**Comment:**

Dear Reviewers,

We sincerely thank you for taking the time to review our work and provide valuable comments. After carefully considering the feedback and discussing the points raised, we have decided to withdraw our submission at this time.

We greatly appreciate the reviewers’ constructive comments, which have offered valuable insights for improving our work. We highly appreciate the questions that have been raised during this rebuttal period and we intend to incorporate changes ahead of the re-submission to the next appropriate venue.

We thank you again for your time and efforts in reviewing our paper.

Best regards,

Authors

**Withdrawal Confirmation:**

I have read and agree with the venue's withdrawal policy on behalf of myself and my co-authors.